# The FANCM-BLM-TOP3A-RMI complex suppresses alternative lengthening of telomeres (ALT)

Robert Lu[1,5], Julienne J. O'Rourke[2,3,5], Alexander P. Sobinoff[1,5], Joshua A.M. Allen[1], Christopher B. Nelson[1], Christopher G. Tomlinson[1], Michael Lee [1], Roger R. Reddel [4], Andrew J. Deans [2,3] & Hilda A. Pickett[1]

The collapse of stalled replication forks is a major driver of genomic instability. Several committed mechanisms exist to resolve replication stress. These pathways are particularly pertinent at telomeres. Cancer cells that use Alternative Lengthening of Telomeres (ALT) display heightened levels of telomere-specific replication stress, and co-opt stalled replication forks as substrates for break-induced telomere synthesis. FANCM is a DNA translocase that can form independent functional interactions with the BLM-TOP3A-RMI (BTR) complex and the Fanconi anemia (FA) core complex. Here, we demonstrate that FANCM depletion provokes ALT activity, evident by increased break-induced telomere synthesis, and the induction of ALT biomarkers. FANCM-mediated attenuation of ALT requires its inherent DNA translocase activity and interaction with the BTR complex, but does not require the FA core complex, indicative of FANCM functioning to restrain excessive ALT activity by ameliorating replication stress at telomeres. Synthetic inhibition of FANCM-BTR complex formation is selectively toxic to ALT cancer cells.

[1] Telomere Length Regulation Unit, Children's Medical Research Institute, Faculty of Medicine and Health, University of Sydney, Westmead 2145 NSW, Australia. [2] Genome Stability Unit, St. Vincent's Institute, 9 Princes St, Fitzroy 3065 VIC, Australia. [3] Department of Medicine (St. Vincent's), University of Melbourne, Parkville 3052 VIC, Australia. [4] Cancer Research Unit, Children's Medical Research Institute, Faculty of Medicine and Health, University of Sydney, Westmead 2145 NSW, Australia. [5] These authors contributed equally: Robert Lu, Julienne J. O'Rourke, Alexander P. Sobinoff. Correspondence and requests for materials should be addressed to A.J.D. (email: adeans@svi.edu.au) or to H.A.P. (email: hpickett@cmri.org.au)

Telomeres are nucleoprotein structures consisting of 5′-TTAGGG-3′ tandem repeats, bound by the protein complex shelterin, that function to protect the ends of linear chromosomes from being recognized as DNA double strand breaks (DSBs)[1]. Their repetitive nature and propensity to form secondary structures such as telomere-loops (t-loops), DNA-RNA hybrids, and G-quadruplexes mean that telomeres are prone to replication stress[2]. Committed mechanisms are therefore in place to remodel stalled replication forks and remove lesions, to facilitate the smooth transition of the replication machinery through to the ends of chromosomes. Numerous proteins contribute to the integrity of telomere replication, including shelterin components[2,3], helicases such as BLM, RTEL1, and SMARCAL1[4–6], and DNA repair proteins such as FANCM and BRCA1[7]. The complex and dynamic interplay of these enzymes is incompletely understood.

Approximately 15% of cancers utilize the ALT pathway of telomere maintenance. ALT cells are characterized by elevated levels of DNA damage compared to mortal or telomerase-positive cells[8,9], indicative of heightened telomeric replication stress in ALT cells. This is attributed to cumulative inadequacies in telomere structural integrity[10]. Frequent or persistent replication fork stalling causes nicks and breaks in the DNA, and it has been hypothesized that the ALT mechanism emanates from stalled replication forks that deteriorate to form DSBs, that then provide the substrate for the engagement of homology-directed repair pathways, culminating in break-induced telomere synthesis[10,11]. ALT telomeres therefore achieve a fine balance between telomere protection and telomere damage and repair activities, and disruption of this balance has the potential to dysregulate the ALT mechanism.

FANCM is an integral factor in the stabilization of stalled replication forks[12]. It contains two DNA binding domains at its N-terminus and C-terminus[13–15], between which are three highly conserved regions (MM1–MM3). The MM1 domain recruits the FA core complex, a multi-subunit ubiquitin ligase that is essential for DNA interstrand crosslink (ICL) repair, while the MM2 domain binds directly to the RMI1–RMI2 subcomplex of BLM-TOP3A-RMI (BTR)[16]. The BTR complex encompasses BLM helicase activity, TOP3A decatenation activity, branch migration, and overall dissolvase activity[17–20], and it has been suggested that FANCM and BTR may cooperate to regress, and thus stabilize, stalled forks[21]. FANCM retention at stalled replication forks is dependent on its interaction with a functional BTR complex, but not with the FA core complex[22]. This suggests FA core complex-dependent and BTR-dependent functions for FANCM. Consistent with these findings, the FA core and BTR complexes bind to FANCM independently, and do not bind to each other, placing FANCM at the helm of these two synergistic pathways[21].

Several FA-associated proteins directly associate with ALT telomeres[23], and have demonstrated roles in facilitating replication through telomere repeat arrays[24,25]. For instance, mono-ubiquitinated FANCD2 has been proposed to promote intramolecular resolution of stalled replication forks in ALT telomeres, thus restraining ALT activity in a BLM-dependent manner[26]. In addition, FANCM, BRCA1, and BLM have been shown to cooperatively resolve replication stress at ALT telomeres. Specifically, this study found that depletion of FANCM resulted in ALT-specific induction of telomeric CHK1 signaling, that BLM, RAD51, and BRCA1 were recruited to ALT telomeres in the absence of FANCM, and that co-depletion of FANCM and BLM, or FANCM and BRCA1, resulted in synthetic lethality of ALT cells[7]. The direct effects of FANCM depletion on ALT activity have not been determined.

Here, we demonstrate that FANCM depletion results in exacerbation of the ALT phenotype, characterized by elevated levels of telomere dysfunction, accumulation of nascent extra-chromosomal telomeric repeat (ECTR) DNA, and increased firing of break-induced telomere synthesis events in the absence of overall telomere length changes, with ultimately detrimental effects to cell viability characterized by attenuated mitotic entry. We identify that the MM2 and DEAH translocase domains of FANCM are primarily responsible for the suppression of excessive ALT activity. The MM2 domain binds to the BTR complex, consistent with the replication fork remodeling capabilities of FANCM-BTR being required for the maintenance of ALT telomere integrity. These data suggest that sites of unresolved replication stress or fork collapse are the direct substrates for ALT-mediated telomere synthesis. Finally, we demonstrate that synthetic disruption of the FANCM-BTR complex by either expression of an MM2 peptide that competitively binds to BTR, or using a small molecule inhibitor of the MM2 domain, can induce ALT-specific loss of cell viability.

## Results

**FANCM depletion induces telomere dysfunction and ALT markers.** ALT telomeres are characterized by an elevated DNA damage response (DDR) compared to mortal and telomerase-positive cells[9]. Such damage is observed as telomere dysfunction-induced foci (TIFs) marked by colocalization of the DNA damage marker γ-H2AX with telomeric DNA. Knockdown of FANCM, using two different siRNAs (Supplementary Figs. 1a, 7), resulted in a significant increase in metaphase-TIFs (meta-TIFs) compared to the scrambled control in U-2 OS ALT cells (Fig. 1a). In contrast to the ALT-specific induction of telomere dysfunction, FANCM depletion induced comparable levels of global DDR signaling[27] in both U-2 OS (ALT) and HeLa (telomerase-positive) cell lines (Supplementary Fig. 1b).

ALT cells characteristically have long and heterogeneous telomere lengths and abundant extrachromosomal telomere repeat (ECTR) DNA, including t-circles and C-circles[28]. One-dimensional gel electrophoresis of isolated terminal restriction fragments (TRFs) followed by hybridization under native conditions revealed both a decrease in the G-rich overhang and a striking increase in low molecular weight single-stranded (ss) C-rich telomeric DNA with FANCM depletion (Fig. 1b; black arrow and red arrow, respectively). This low molecular weight species of DNA was also detected following separation of undigested genomic DNA (Supplementary Fig. 1c), indicative of it being extrachromosomal in origin. Despite these observations, no change in mean telomere length was observed (Fig. 1b). These effects were observed following FANCM depletion using both siRNAs, with the most marked effects being seen with siFANCM2. Consequently, siFANCM2 was used for the subsequent experiments. Separation of TRFs by two-dimensional gel electrophoresis identified an increase in extrachromosomal t-circles, which characteristically resolve as an arc above the linear telomeric DNA arc, following FANCM depletion (Fig. 1c; blue arrows). The low molecular weight ss C-rich telomeric DNA ran as a distinct and separate arc below both the t-circle and linear telomeric DNA arcs in FANCM depleted cells (Fig. 1c; red arrows).

FANCM depletion resulted in a striking increase in C-circles, detected by rolling circle amplification (Fig. 1d and Supplementary Fig. 1d). This increase in amplified C-circles corresponded with the low molecular weight ss ECTR C-rich telomeric DNA identified by one-dimensional and two-dimensional gel electrophoresis (Fig. 1b, c), consistent with this species of DNA being C-circles. This is the first reported visualization of C-circles by gel electrophoresis, demonstrating that t-circles and C-circles resolve as distinct DNA species. Quantitation of ALT-associated PML

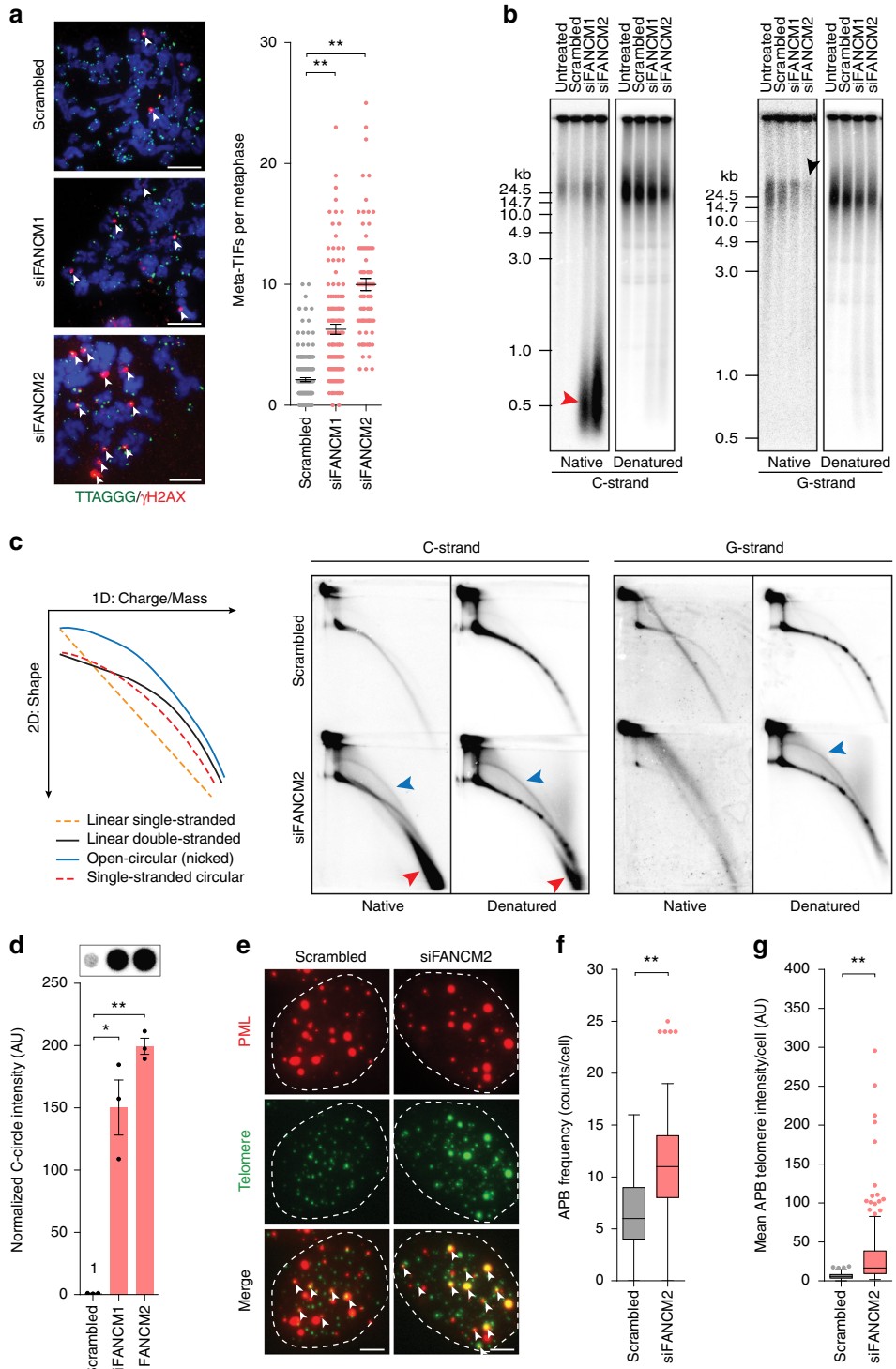

**Fig. 1** FANCM depletion results in telomere dysfunction and increased ALT activity. **a** Representative images of telomere (green) and γ-H2AX (red) colocalizations on metaphase spreads (meta-TIFs) in U-2 OS cells with or without FANCM depletion (left panel). Meta-TIFs are indicated by white arrows. Scale bars are 5 μm. Quantitation of TIFs (right panel). Scatterplot bars represent the mean ± SEM. Out of three experiments, $n \geq 81$ metaphases scored per treatment, $**p < 0.005$, Mann–Whitney test. **b** Native and denatured TRF analysis of U-2 OS cells with or without FANCM depletion. Colored arrows correspond to telomeric DNA species depicted in **c**. **c** Schematic of migration patterns for indicated telomeric species separated by two-dimensional gel electrophoresis (left panel). Two-dimensional TRF analysis of U-2 OS cells with or without FANCM depletion, hybridized under native and denatured conditions to detect the telomeric C-strand and G-strand (center and right panels). Colored arrows correspond to telomeric DNA species depicted in schematic. **d** Representative dot blot and quantitation of C-circle assays in U-2 OS cells with or without FANCM depletion. C-circle levels were normalized to the mean of scrambled control. Error bars represent the mean ± SEM from $n = 3$ experiments, $*p < 0.05$, $**p < 0.005$, Student's $t$-test. **e** Representative images of telomere (green) and PML (red) colocalizations (APBs) in FANCM-depleted U-2 OS cells. APBs are indicated by white arrows. Scale bars are 5 μm. Tukey box plots of **f** APB frequency and **g** mean APB telomere foci intensity per cell. Out of three experiments, $n \geq 150$ cells scored per treatment, $*p < 0.05$, $**p < 0.005$, Mann–Whitney test

bodies (APBs), in which telomeric DNA and telomere-associated proteins colocalize with PML protein, revealed a significant increase in the number of APBs following FANCM depletion in U-2 OS and IIICF/c cells, but no change in GM847 and Saos-2 cells (Fig. 1e, f and Supplementary Fig. 1e). Interestingly, a significant increase in the intensity of the telomeric signal within APBs was also observed in all ALT cell lines analyzed (Fig. 1e, g and Supplementary Fig. 1f), indicative of increased telomeric DNA accumulation or telomere clustering in APBs in response to FANCM depletion. Together, these data demonstrate a striking induction of ALT phenotypes, but no overall telomere lengthening, following transient loss of FANCM. These observations were specific to ALT cell lines (U-2 OS, IIICF/c, GM847, and Saos-2) and independent of p53 status, while no induction of ALT characteristics was observed in response to FANCM depletion in telomerase-positive cell lines (HeLa, HeLa 1.2.11, and HCT116) (Supplementary Fig. 1), indicative of the induction of ALT phenotypes being attributable to ALT-specific telomere dysfunction.

**FANCM depletion promotes break-induced telomere synthesis**. ALT involves a mechanism of conservative break-induced telomere synthesis that is analogous to break-induced replication (BIR) in yeast[29], and is dependent on the POLD3 subunit of Polδ[10,11]. We identified a significant increase in POLD3 recruitment to telomeres following FANCM depletion (Fig. 2a). This coincided with an increase in total nascent telomere repeat generation (Fig. 2b). Nascent telomeric DNA localized with APBs (Fig. 2c), and contributed to the dramatic induction of C-circles generated in response to FANCM depletion (Fig. 2d).

It has previously been shown that break-induced telomere synthesis is dependent on BLM[10], and that both RAD51-dependent and RAD51-independent pathways of telomere extension can exist[11,30,31]. These pathways are reminiscent of Type I (Rad51-dependent) and Type II (Rad51-independent) telomerase-null *Saccharomyces cerevisiae* survivors, which require Rad52 and BLM homolog Sgs1[32]. To characterize the involvement of these proteins in the context of FANCM knockdown, we co-depleted FANCM and either POLD3, BLM, RAD51, and RAD52 (Supplementary Figs. 2a and 7). Interestingly, FANCM depletion resulted in a concomitant decrease in protein levels of POLD3, BLM, and RAD51 (Supplementary Fig. 2a, b), which was significant for RAD51, suggesting that FANCM coregulates these proteins. It is unlikely that this coregulation contributes to the exacerbated ALT phenotype observed, as independent depletion of POLD3, BLM, or RAD51 causes subtle or antagonistic effects to ALT activity[10], compared to that seen with FANCM depletion. Co-depletion experiments showed that the elevated levels of C-circles detected by both the C-circle assay and by TRF analysis following FANCM depletion were dependent on POLD3 and BLM, and partially dependent on RAD51 and RAD52 (Fig. 3a and Supplementary Fig. 2c). Similarly, the increased number and intensity of APBs in response to FANCM depletion was dependent on POLD3 and BLM, and partially dependent on RAD51 and RAD52 (Fig. 3b, c).

To directly measure the frequency and length of telomere synthesis events, we used single molecule analysis of telomeres (SMAT) on CldU-incorporated DNA fibers[10]. FANCM-depletion resulted in a significant increase in the number of telomere extension events, while the length of the extension products remained unchanged (Fig. 3d). This increase was predominantly dependent on POLD3, BLM, and RAD52 (Fig. 3d). Overall, these data demonstrate that FANCM depletion results in an increase in POLD3- and BLM-mediated break-induced telomere synthesis at ALT telomeres, and coincides with a rapid induction of

nascent ECTRs that are predominantly ss and C-rich. These data indicate a stronger reliance on RAD52 than RAD51, but support a role for both proteins in break-induced telomere synthesis.

**The MM2 domain of FANCM is required to attenuate ALT activity**. The multiple functional domains of FANCM have been well characterized. Specifically, the PIP domain interacts with proliferating cell nuclear antigen (PCNA) via a conserved PIP-box sequence[33]. The conserved DEAH domain has ATP-dependent DNA-remodeling translocase activity, and binds to replication forks and DNA repair intermediates to promote displacement and annealing of nascent and parental DNA strands[13,14]. The Major Histone Fold 1 and 2 (MHF1/2) heterotetramer binds to the MHF1/2 interacting domain (MID), and is an obligate cofactor that targets FANCM to DNA branch points to facilitate replication fork traversal[34–37]. The MM1 domain recruits the FA core complex to ICL sites; the MM2 domain interacts with the BTR complex to coordinate the stabilization of replication forks; and the MM3 domain has no known function. A key site at S1045 is phosphorylated upon genotoxic stress and is required for efficient ATR-CHK1 checkpoint activation[38]. The ERCC4 endonuclease domain and the Helix-hairpin-Helix (HhH) motif recognize ss DNA gaps and lesions in DNA present at ICL sites, and facilitate heterodimerization with FAAP24, another obligate FANCM cofactor[13,27,39,40].

It has been shown elsewhere that FANCM directly associates with telomeric DNA[7,41]. To determine the specific functional activity of FANCM required to suppress the ALT phenotype, we established a panel of ten mutant lentiviral constructs, in which each identified domain was disrupted (Fig. 4a). The FF>AA mutant has a double substitution of phenylalanine for alanine (FANCM$^{F1232A/F1236A}$) within the MM2 domain, and has previously been characterized to substantially disrupt BTR binding[21]. The MID domain mutant is substituted at V749G/H751G to disrupt interaction with the MHF1/2 heterotetramer interface[36]. These constructs, including wild-type FANCM, were stably transduced into U-2 OS cells, and exogenous expression of the FANCM mutants confirmed by western blot analysis (Supplementary Figs. 3a and 7). Stable overexpression of FANCM resulted in a significant decrease in both telomere dysfunction (Fig. 4b) and the number of fragile telomeres (Fig. 4c). Most of the mutants also suppressed the telomere DDR and telomere fragility (Fig. 4b, c). The exceptions were the MM2, FF>AA and MID domain mutants that significantly increased both telomere dysfunction and telomere fragility, and the K117R mutant in which the DEAH domain is disrupted, that failed to suppress the phenotypes (Fig. 4b, c).

We then examined the effects of FANCM mutant overexpression on ECTR generation and ALT activity. Consistent with stable overexpression of wild-type FANCM, overexpression of most FANCM mutants resulted in a significant decrease in C-circle generation (Fig. 4d). The exceptions were the K117R, ERCC4, and HhH domain mutants that did not cause a change in C-circle levels, and the MM2 and FF>AA domain mutants that caused an approximate two-fold increase in C-circles compared to vector control (Fig. 4d). No changes in mean telomere length were observed following wild-type or mutant FANCM overexpression; however, the previously identified smear of low molecular weight ss C-rich telomeric DNA was observed in cells overexpressing the K117R, MM2, and FF>AA domain mutants (Supplementary Fig. 3b). These changes coincided with a significant increase in both the number of APBs and the intensity of telomeric signal within APBs following overexpression of the K117R, MM2 and FF>AA mutants (Supplementary Fig. 3c).

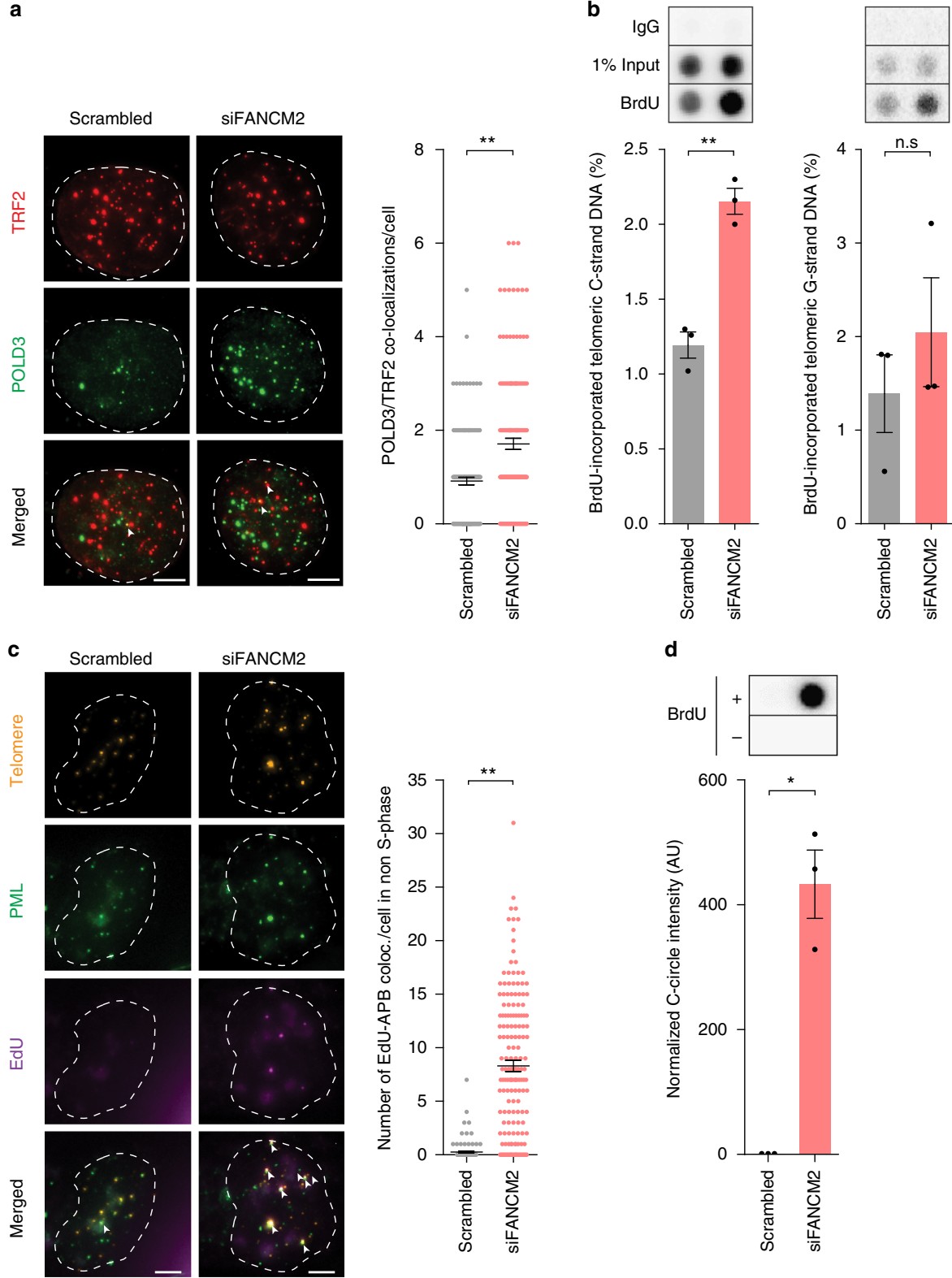

More subtle differences in the number and intensity of APBs were observed following overexpression of some of the other mutants, indicative of the underlying complexity of APB formation and function.

We then measured the number and length of telomere extension events to determine whether overexpression of FANCM mutants had a direct impact on telomere synthesis. A significant increase in the number of extension events was observed in response to overexpression of the MM2 and FF>AA domain mutants (Fig. 4e), while the length of extension events remained relatively stable following overexpression of wild-type FANCM and the other mutants. These data show that wild-type FANCM suppresses telomere replication stress, but does not directly impact the length of telomere extension events. Overall, we demonstrate the functional significance of the replication fork remodeling and restart activities of FANCM, provided by its

**Fig. 2** FANCM depletion results in the generation of nascent telomeric DNA. **a** Representative images of TRF2 (red) and POLD3 (green) colocalizations in U-2 OS cells with or without FANCM depletion (left panel). Colocalizations are indicated by white arrows. Scale bars are 5 μm. Quantitation of colocalizations (right panel). Scatterplot bars represent the mean ± SEM. Out of three experiments, n = 150 cells scored per treatment, *p < 0.05, **p < 0.005, Mann–Whitney test. **b** Representative dot blot and quantitation of nascent telomeric C-strand (left panel) and G-strand (right panel) DNA following BrdU incorporation and immunoprecipitation in U-2 OS cells with or without FANCM depletion. Nascent telomeric content was normalized to serial dilution of input DNA. Error bars represent mean ± SEM from n = 3 experiments, **p < 0.005, n.s. = non-significant, Student's t-test. **c** Representative images of telomere (orange), PML (green) and EdU (violet) colocalizations (EdU-APB) in U-2 OS cells with or without FANCM depletion (left panel). Scale bars are 5 μm. Quantitation of colocalizations (right panel). Scatterplot bars represent the mean ± SEM. Out of three experiments, n ≥ 122 non-S-phase cells scored per treatment, **p < 0.005, Mann–Whitney test. **d** Representative dot blot and quantitation of C-circle assays following BrdU incorporation and immunoprecipitation in U-2 OS cells with or without FANCM depletion. C-circles were normalized to the mean of scrambled control. Error bars represent mean ± SEM out of n = 3 experiments, **p < 0.005, Student's t-test

---

ATP-dependent translocase activity (DEAH domain) and its BTR binding capability (MM2 domain), in regulating ALT activity. The data also indicate dominant-negative effects following overexpression of the MM2 domain mutants.

**Disruption of the FANCM-BTR complex inhibits ALT cell viability.** It has previously been shown that FANCM depletion induced replication stress at ALT telomeres, but did not affect cell viability[7]. In contrast, throughout the course of our study, we consistently observed a paucity of mitotic cells following FANCM depletion. To determine the effects of FANCM depletion on cell cycle progression, we used cell cycle analysis and live cell imaging. We observed an accumulation of cells in G2/M in FANCM depleted U-2 OS cells compared to FANCM depleted HeLa cells (Supplementary Fig. 4a, b). Both U-2 OS and HeLa cells showed delayed cycling speed following FANCM depletion, as evident by an increase in interphase duration (Supplementary Fig. 4c, d). U-2 OS cells depleted of FANCM exhibited cell cycle attenuation, with more than a third of cells failing to enter mitosis over the 48 h observation window, while the effect of FANCM depletion on mitotic entry in HeLa cells was considerably less severe (Supplementary Fig. 4c, d). No changes in mitotic outcome (aberrant mitosis or mitotic death) were apparent following FANCM depletion, consistent with cells arresting prior to mitosis (Supplementary Fig. 4e, f). It is likely that cells capable of progressing through to mitosis had evaded critical levels of FANCM knockdown. These data demonstrate ALT cells are hypersensitive to replication stress caused by FANCM depletion.

To further evaluate the effect of FANCM perturbation on cell survival, we utilized Project Achilles, an initiative to identify and catalog gene essentiality across cancer cell lines[42]. We compared the gene dependency scores for FANCM following CRISPR-Cas9-mediated knockout. Gene dependency scores were calculated using CERES[43] and indicate the likelihood that FANCM is essential in the cell line. A lower gene dependency score is indicative of a higher likelihood that the gene is essential, with −1 representing the median score for all pan-cancer essential genes. Overall, across the panel of cell lines, FANCM did not appear to be essential for cell viability (Supplementary Fig. 5). However, for the subset of ALT cell lines that were identified, the majority (4/5) clustered around a gene dependency score of −1 (Supplementary Fig. 5). These data support an essential role for FANCM in ALT cancer cell viability.

To explore the possibility of specifically targeting ALT cell viability through FANCM, we used two approaches to disrupt the FANCM-BTR complex. First, we used a chemically synthesized peptide corresponding to the highly conserved 28 amino acids of the MM2 domain, that has previously been shown to have similar binding affinity to the BTR complex as full length FANCM[21]. Addition of the MM2 peptide to cellular lysates resulted in dose-dependent inhibition of FANCM-BTR complex formation, whilst

addition of an MM2 peptide containing the FF>AA mutant that does not bind to BTR was unable to inhibit complex formation (Supplementary Figs. 6a, b and 8). To investigate the effects of FANCM-BTR complex disruption in ALT cells, the MM2 peptide was rendered functionally dependent on tamoxifen by fusion to the C-terminus of the estrogen receptor (MM2-ER) (Fig. 5a). The ability of the MM2-ER fusion protein to inhibit FANCM-BTR complex formation was tested by immunoprecipitation with either FANCM or ER. In the absence of tamoxifen, TOP3A and RMI1 components of the BTR complex were immunoprecipitated by FANCM, but not by ER (Fig. 5b and Supplementary Fig. 8). Addition of tamoxifen resulted in activation of the MM2-ER fusion protein, detected by immunoprecipitation of TOP3A and RMI1 with ER, but not with FANCM. Activation of the control FF>AA mutant MM2-ER fusion protein was unable to sequester the BTR complex away from FANCM (Fig. 5b).

To confirm the genomic efficacy of FANCM-BTR complex disruption, we quantitated sister chromatid exchanges (SCEs). It has previously been shown that FANCM depletion results in increased SCE formation[21]. We identified an increase in the frequency of SCEs in response to activation of the MM2-ER fusion protein, but not the FF>AA mutant (Supplementary Fig. 6c, d), consistent with effective disruption of the FANCM-BTR complex in both ALT and telomerase-positive cell lines. Activation of the MM2-ER fusion protein resulted in a significant increase in TIFs in U-2 OS cells, while a small but significant decrease in TIFs was observed following activation of the FF>AA mutant fusion protein (Fig. 5c). Activation of the MM2-ER fusion protein, but not the FF>AA mutant, also induced C-circles (Fig. 5d), indicative of enhanced replication stress at ALT telomeres leading to increased ALT activity. Clonogenic survival assays revealed a 10–20-fold decrease in the survival of U-2 OS, GM847, and Saos-2 ALT cells following activation of the MM2-ER fusion protein, while activation of the FF>AA mutant MM2-ER fusion protein did not affect cell survival (Fig. 5e). No impact on cell survival was observed with activation of either the wild-type MM2-ER or FF>AA mutant MM2-ER fusion proteins in the telomerase positive cell lines HeLa and HCT116 (Fig. 5e).

The second approach involved treating cells with increasing concentrations of PIP-199, a small molecule inhibitor of the MM2-RMI interaction within FANCM-BTR[44]. We observed a moderate increase in C-circles in the U-2 OS, GM847, and Saos-2 ALT cell lines (Fig. 5f), consistent with PIP-199-mediated disruption of the FANCM-BTR complex. Clonogenic assays identified hypersensitivity of ALT cells to PIP-199 compared to telomerase-positive cells (Fig. 5g). These experiments employ two independent approaches to disrupt the critical binding interaction between FANCM and the BTR complex, and demonstrate that inhibition of the FANCM-BTR complex causes replication stress and elevated C-circles. Importantly, FANCM-BTR complex inhibition can be used to selectively suppress the growth and viability of ALT cancer cells.

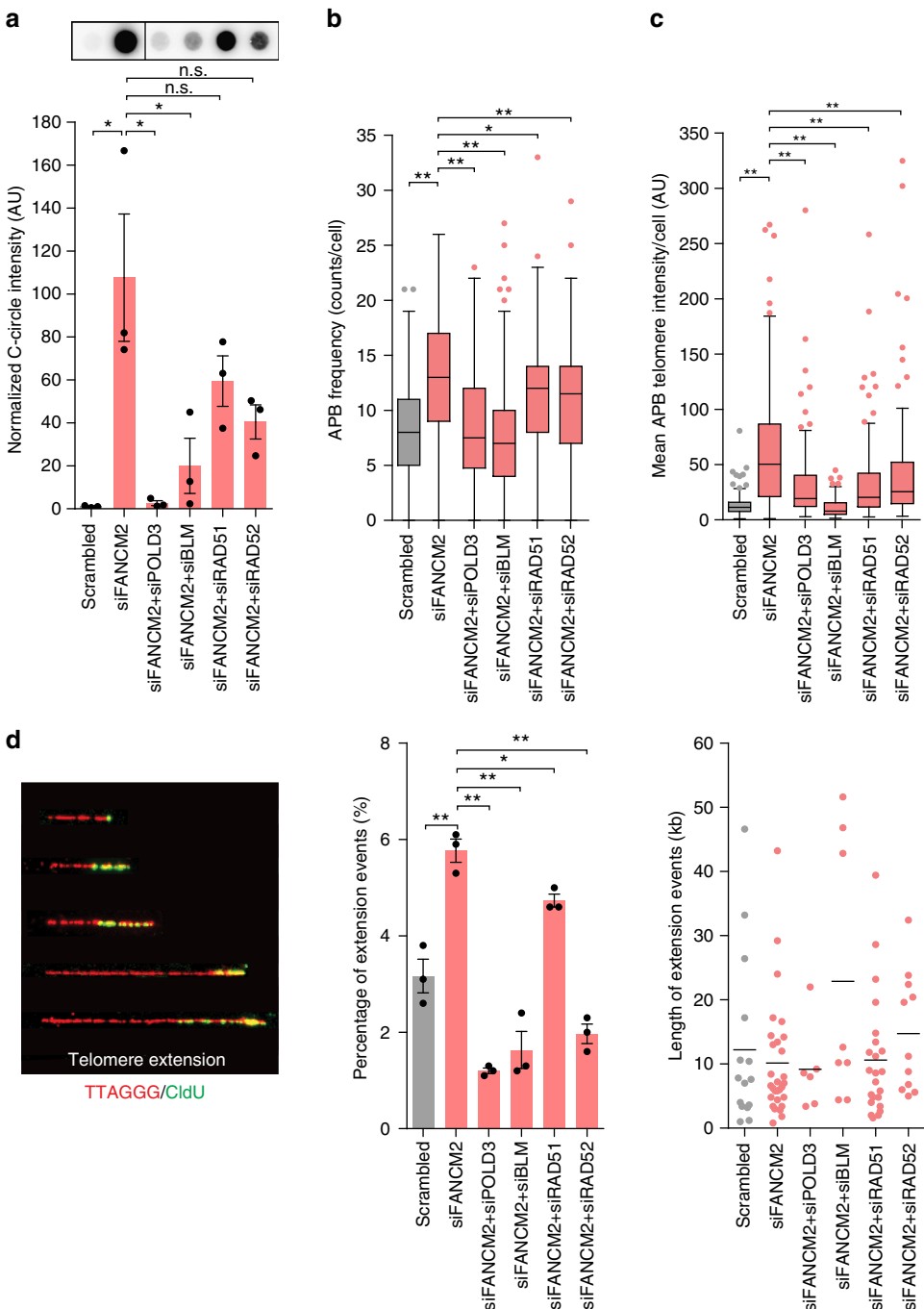

**Fig. 3** FANCM depletion results in increased break-induced telomere synthesis. **a** Representative dot blots and quantitation of C-circles in U-2 OS cells co-depleted of FANCM and either POLD3, BLM, RAD51, or RAD52. C-circles were normalized to the mean of scrambled control. Error bars represent mean ± SEM from $n = 3$ experiments, *$p < 0.05$, n.s. = non-significant, Student's $t$-test. Quantitations of **b** APB frequency and **c** mean APB telomere foci intensity in U-2 OS cells co-depleted of FANCM and either POLD3, BLM, RAD51, or RAD52. Scatterplot bars represent the mean ± SEM. Out of three experiments, $n = 150$ cells scored per treatment, *$p < 0.05$, **$p < 0.005$, Mann–Whitney test. **d** Examples of telomere extension fibers (red) scored after CldU incorporation (green) (left panel). Quantitation of the number and length of telomere extension events in U-2 OS cells co-depleted of FANCM and either POLD3, BLM, RAD51 or RAD52 (center and right panels). Error bars represent mean ± SEM of $n \geq 350$ fibers out of three experiments, *$p < 0.05$, **$p < 0.005$, Student's $t$-test

## Discussion

FANCM is a non-essential gene for eukaryote development, and siRNA-mediated knockdown, gene targeted deletion or inherited somatic dysfunction of FANCM is tolerated in humans, mice, flies and plants[21,45–47], albeit with phenotypes of reduced fertility and tumor predisposition[45,48,49]. In the context of our study, we were unable to obtain FANCM knockout clones or achieve

stable FANCM knockdown in ALT-positive cell lines. Consistent with ALT-specific intolerance to FANCM depletion, we observed reduced cell viability in ALT cell lines following siRNA-mediated knockdown of FANCM. Our data indicate that FANCM depletion is detrimental to ALT cells, and that this toxicity is mediated by telomere dysfunction, which promotes unrestrained or exaggerated ALT activity, culminating in G2/M cell cycle arrest.

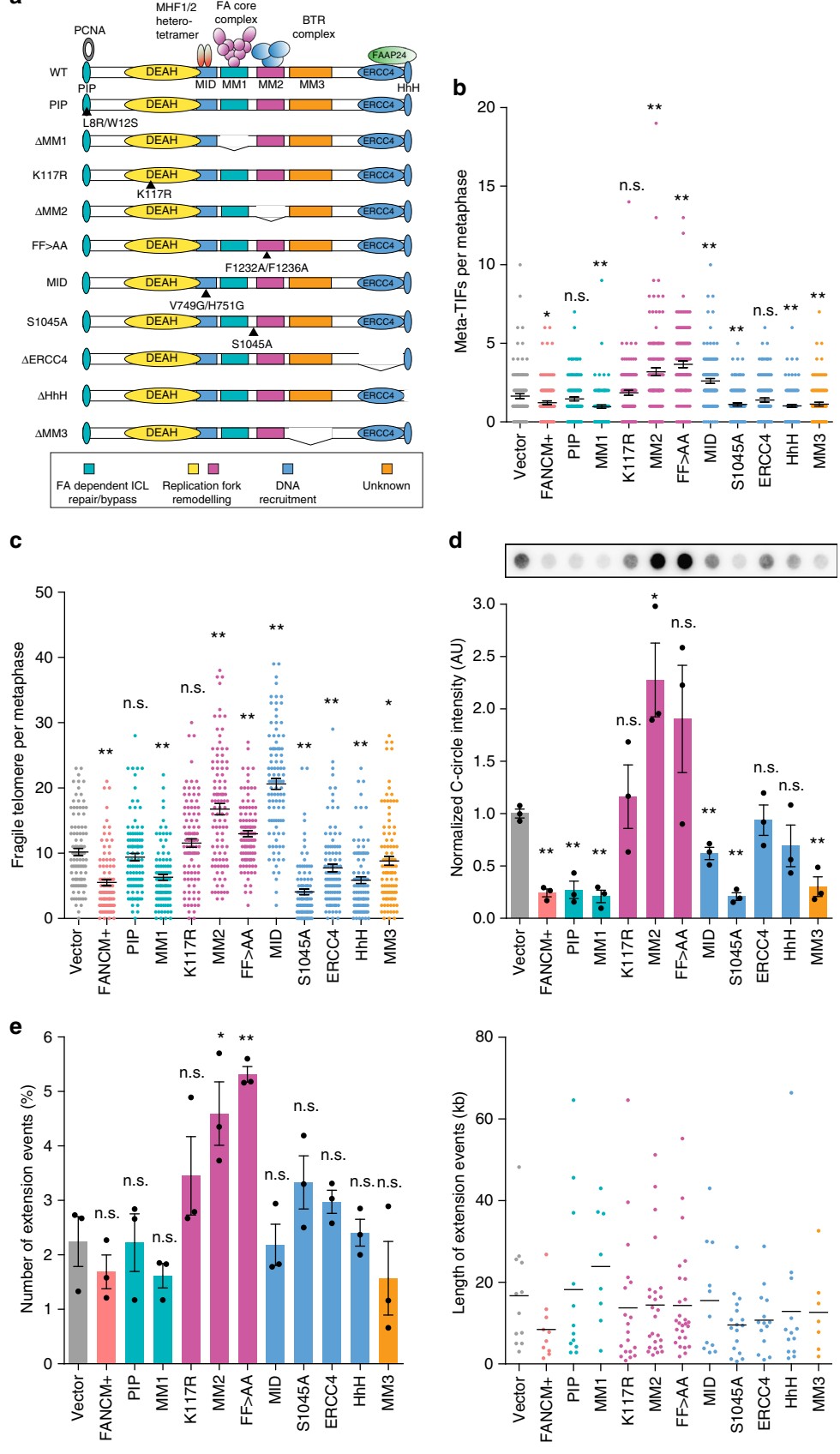

**Fig. 4** ALT activity is attenuated by the replication fork remodeling capabilities of FANCM. **a** Schematic of FANCM domain interactions, mutations or deletions. For clarity, domains have been color-coded by functional role (bottom panel). **b** Quantitation of metaphase-TIFs in U-2 OS cells stably overexpressing wild-type (FANCM+) or FANCM mutants. Scatterplot bars represent the mean ± SEM. Out of three experiments, $n \geq 110$ metaphases scored for each mutant, $*p < 0.05$, $**p < 0.005$, Mann–Whitney test. **c** Quantitation of fragile telomeres in U-2 OS cells stably overexpressing wild-type (FANCM+) or FANCM mutants. Scatterplot bars represent the mean ± SEM. Out of three experiments, $n > 100$ metaphases scored for each mutant, $*p < 0.05$, $**p < 0.005$, Mann–Whitney test. **d** Representative dot blots and quantitation of C-circle assays in U-2 OS cells stably overexpressing wild-type (FANCM+) or FANCM domain mutants. C-circles were normalized to the mean of vector control. Error bars represent mean ± SEM from $n = 3$ experiments, $*p < 0.05$, $**p < 0.005$, Student's $t$-test. **e** Single molecule analysis of telomeric extension events (left panel) and length of extension events (right panel) in U-2 OS cells stably overexpressing wild-type (FANCM+) or FANCM domain mutants. Error bars represent mean ± SEM of $n \geq 350$ fibers out of three experiments, $*p < 0.05$, $**p < 0.005$, Student's $t$-test

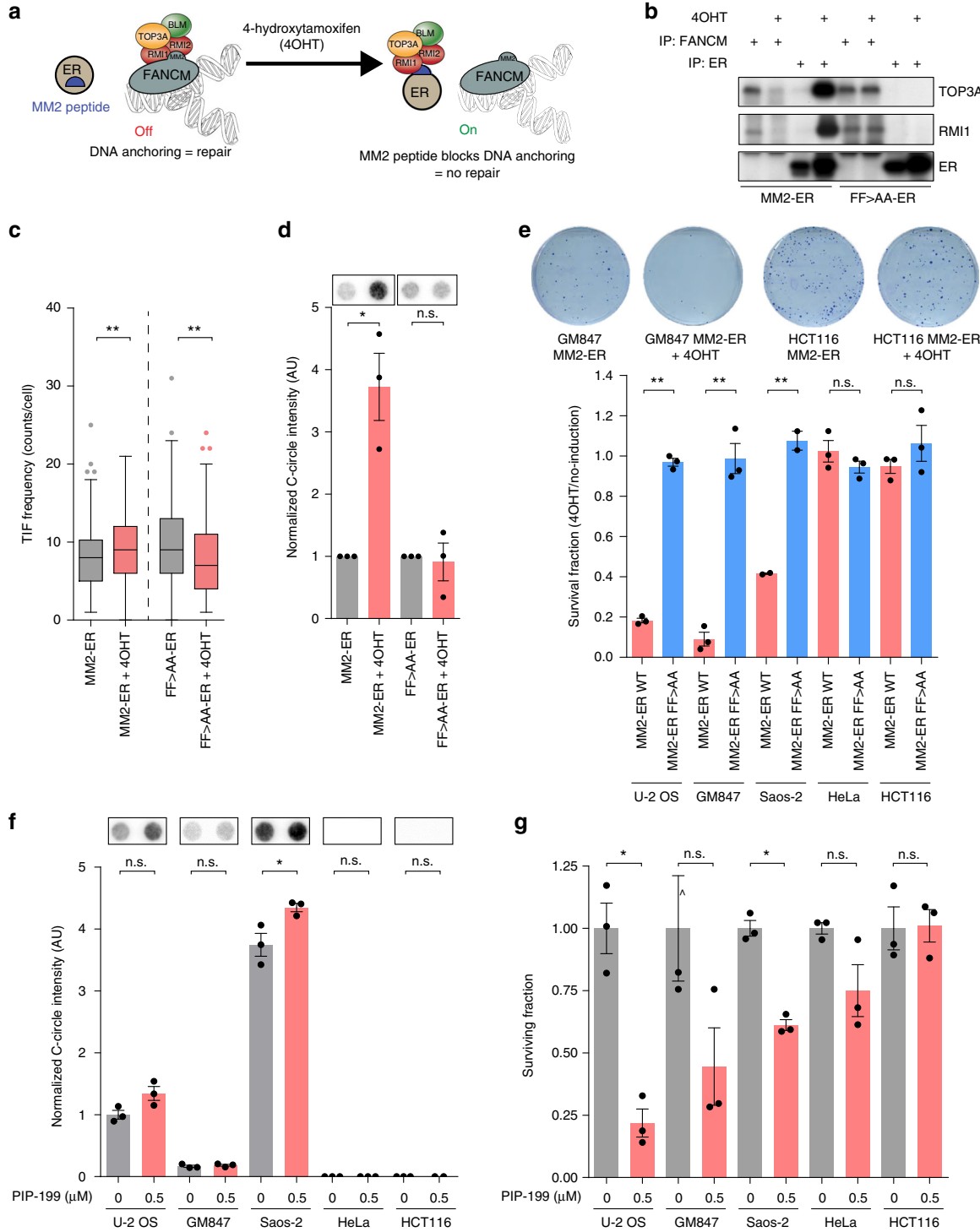

**Fig. 5** Inhibition of the FANCM-BTR complex results in loss of ALT cell viability. **a** Schematic of MM2- tamoxifen (4OHT)-inducible ER fusion protein-mediated inhibition of the FANCM-BTR complex interaction. **b** Immunoprecipitation confirming the transfer of TOP3A and RMI1 from FANCM to the decoy MM2-ER fusion protein following 4OHT activation in U-2 OS cells. Transfer of complex components is not seen with the FF>AA mutant MM2-ER protein (FF>AA-ER). **c** Tukey boxplots of TIFs in U-2 OS cells expressing MM2-ER or FF>AA-ER fusion proteins in the presence or absence of 4OHT. Out of three experiments, $n = 150$ cells scored per treatment, $**p < 0.005$, Mann–Whitney test. **d** Representative dot blots and quantitation of C-circle assays in U-2 OS cells expressing MM2-ER or FF>AA-ER fusion proteins in the presence or absence of 4OHT. C-circles were normalized to non-induced control. Error bars represent mean ± SEM from $n = 3$ experiments, $*p < 0.05$, one-sample $t$-test. **e** Representative colony formation assays of GM847 and HCT116 cells expressing the MM2-ER fusion protein in the presence or absence of 4OHT (top panel). Quantitation of the surviving fraction of colonies for ALT (U-2 OS, GM847, and Saos-2) and telomerase-positive (HeLa and HCT116) cell lines expressing MM2-ER or FF>AA-ER fusion proteins (bottom panel). Colony counts were normalized to non-induced controls. Error bars represent mean ± SEM from $n = 3$ experiments, $**p < 0.005$, n.s. = non-significant, Student's $t$-test. **f** Representative dot blots and quantitation of C-circle assays from U-2 OS, GM847, Saos-2, HeLa, and HCT116 cells treated with 0.5 µM PIP-199 or vehicle control (DMSO) for 72 h. C-circles were normalized to a reference sample control. Error bars represent mean ± SEM from $n = 3$ experiments, $*p < 0.05$, n.s. = non-significant, Student's $t$-test. **g** Quantitation of the surviving fraction of colonies from U-2 OS, GM847, Saos-2, HeLa, and HCT116 cells treated with PIP-199. Colony counts were normalized to DMSO controls. ^Denotes a datapoint (1.42) exceeding the visible axis. Error bars represent mean ± SEM from $n = 3$ experiments, $*p < 0.05$, n.s. = non-significant, Student's $t$-test

Despite the detrimental effects of exacerbated ALT activity, our data also highlight the positive contribution replication stress response inadequacies have in promoting ALT activity.

BIR functions to repair one-ended DSBs through strand invasion and D-loop or bubble migration of an unresolved Holliday junction that displaces the newly synthesized strand, allowing asynchronous leading strand synthesis followed by lagging strand fill-in[50]. The mechanism of break-induced telomere synthesis in ALT cells is analogous to BIR[10,11]. Here, we provide genetic evidence that FANCM attenuates break-induced telomere synthesis in ALT cells. Specifically, FANCM depletion resulted in exacerbation of the ALT phenotype, visualized by elevated telomere dysfunction, increased firing of break-induced telomere synthesis events, and rapid induction of nascent ECTR species, which accumulate in APBs, while exogenous overexpression of FANCM suppressed telomere replication stress, but did not impact ALT-mediated telomere synthesis.

No changes in mean telomere length were observed during transient FANCM depletion; however, the short timeline may preclude the detection of telomere length changes by TRF analysis. These data also suggest that ECTRs are a by-product, rather than a substrate for break-induced telomere synthesis, and that ECTR generation may counteract telomere extension. The exaggerated ALT phenotype observed following FANCM depletion was dependent on POLD3 and BLM, consistent with BIR in human cells[51,52], while partial dependence on RAD51 and RAD52 was observed, implicating the existence of either a single mechanism with RAD51 and RAD52 interdependence, or separate pathways with reliance on either RAD51 or RAD52. This is consistent with a recent publication suggesting that ALT is mediated by two distinct BIR pathways, one dependent on RAD52 and the other independent of RAD52[53]. Induction of all aspects of the ALT phenotype following FANCM depletion implicates FANCM in suppressing an early event in the ALT pathway.

To determine the mechanism by which FANCM regulates break-induced telomere synthesis at ALT telomeres, we conducted a detailed screen of FANCM functional mutants. Our data indicate that two specific domains in FANCM suppress ALT. First, the MM2 domain that is required for the interaction between FANCM and the BTR complex. Second, the DEAH helicase domain that provides the ATP-dependent DNA remodeling translocase activity of FANCM. The combined activities of these two domains facilitate the reversal and restart of stalled replication forks[13,22,27].

Overexpression of two different BTR-binding deficient mutants of FANCM (FF>AA and ΔMM2) or the translocase defective (K117R) mutant resulted in elevated ALT phenotypic markers,

including increased telomere dysfunction and fragility, increased C-circles, and a greater proportion of telomere synthesis events. Our results are reminiscent of observations made in yeast, in which Sgs1 (BLM homolog) and Mph1 (FANCM homolog) positively and negatively regulate BIR, respectively[54,55]. In vertebrates, these two enzymes have acquired an integrated function through the acquisition of RMI2, which bridges the BTR complex to the MM2 domain in FANCM[16,21]. Our data indicate that the MM2 domain mutants elicit a dominant-negative effect on ALT activity, potentially by out-competing DNA-bound endogenous FANCM, and failing to recruit the BTR complex.

The MM1 domain mutant behaved similarly to wild-type FANCM, suppressing all aspects of the ALT phenotype. This demonstrates that FANCM function at ALT telomeres is independent of the FA-core complex. Consistent with this observation, a recent report found that FANCM functioned independently of the FA-core complex in the protection of common fragile sites[56]; however, this study did not investigate the involvement of BLM. A separate study found that co-depletion of FANCM and BLM greatly amplified tandem duplication events through a replication restart bypass mechanism[57]. The ERCC4 and HhH domain mutants also did not increase ALT activity, and failed to suppress the ALT phenotype to the extent seen with wild-type FANCM overexpression. This is consistent with their ICL-specific role in facilitating FANCM binding to FAAP24 at chromatin, and activating the ATR-CHK1 checkpoint in response to ICL-lesions[40]. These data demonstrate that FANCM functions in a BTR-dependent and FA-core complex-independent manner to resolve replication stress at ALT telomeres. Our study does not directly address the type of replication stress encountered by FANCM in the context of ALT, but our data, in the context of previous findings, implicate R-loops[58,59], G-quadruplexes and other telomere secondary structures, rather than ICLs.

Replication fork reversal involves the remodeling of stalled replication forks into four-way junctions to enable the removal or bypass of genetic lesions prior to replication restart[60]. We propose that FANCM fulfils a critical protective function at ALT telomeres by reversing and remodeling stalled replication forks to facilitate smooth transition of the replication machinery (Fig. 6). This is supported by previous observations in which depletion of SMARCAL1, an ATP-driven annealing helicase that mediates fork reversal by annealing nascent DNA strands[61,62], resulted in induction of ALT phenotypes, including TIFs, telomeric signal within APBs, and C-circles[6]. These effects were noticeably milder than those identified here in response to FANCM depletion, and may reflect differences in the underlying fork remodeling mechanism, preference for a particular structure of replication intermediate, or the type of genetic lesion encountered[60].

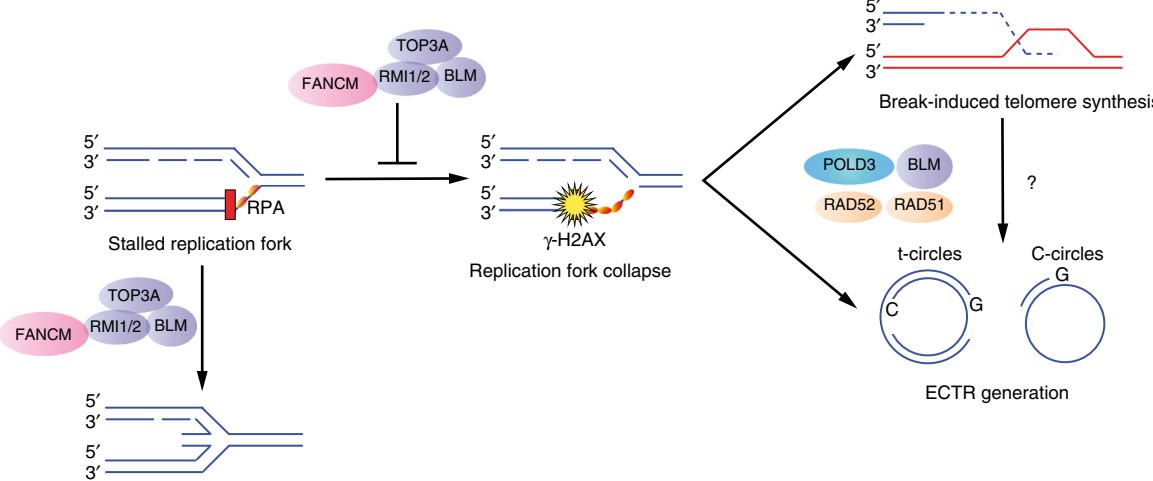

**Fig. 6** Schematic of proposed model of FANCM-mediated ALT suppression. FANCM functions to reverse and remodel stalled replication forks that predominate in ALT telomeres. In the absence of FANCM, or through disruption of the FANCM-BTR complex, stalled forks deteriorate into double strand breaks, which provide the substrate for break-induced telomere synthesis events and the concomitant production of nascent ECTRs

Our data identified rapid induction of nascent telomeric DNA species, including abundant ss C-rich ECTRs, following FANCM depletion. The precise origin of this DNA is unclear. It is possible that persistent stalled forks caused by FANCM depletion provoke unscheduled origin firing, which is responsible for the generation of nascent ss DNA. Nascent ss DNA is typically bound by RPA, but exhaustion of the cellular pool of RPA, in instances of excessive ss DNA production, can further exacerbate DSB generation[63]. It is then possible that in the absence of FANCM, BLM independently localizes at internal DSBs within ALT telomeres[7], and promotes long-range end resection, thus generating ss DNA species[64,65]. This may explain why, in the absence of both FANCM and BLM, both ss ECTR generation and break-induced telomere synthesis events are impeded.

Several previous studies have suggested that inhibition of FANCM function could be used as a mechanism of cancer-specific cell killing[7,16,56]. Our data demonstrate a particularly potent strategy for killing ALT cancer cells, through targeting the FANCM-BTR interaction. While the tamoxifen-inducible ER system has previously been used to directly activate enzymes[66], we used ER-fusion to regulate sequestration of the BTR complex away from FANCM. Using this toggle, we have demonstrated that fusion of the MM2 domain to ER creates a dominant negative switch that can be activated by tamoxifen to phenocopy FANCM depletion.

The structure of the MM2 peptide bound to RMI1:RMI2 reveals a hydrophobic packing arrangement mediated by several key phenylalanine residues. The high affinity ($K_d \sim 5\,nM$) of interaction is mediated by just 9 amino acids within MM2, which extend across a groove formed in a surface of the OB-folds in the RMI1 and RMI2 heterodimer[16]. A high throughput drug discovery screen was used to identify PIP-199 as an inhibitor of this interaction in vitro[44]. Our data demonstrate that PIP-199 similarly phenocopies FANCM depletion and that further development of similar FANCM-BTR interaction inhibitors may provide an effective and selective therapeutic strategy to target ALT cancers. Our work also suggests that the integrity of the BIR pathway is critical to the survival of ALT cells. Members of the BTR complex, POLD3, or other BIR-specific factors could also represent promising targets for ALT-specific cancer therapeutics.

In summary, our study defines the mechanism by which FANCM regulates ALT activity. Specifically, FANCM functions

in a BTR-dependent, FA-core complex-independent manner, to resolve replication stress that arises spontaneously within telomeres. ALT telomeres possess inherent structural aberrations that predispose them to replication defects, and are therefore hypersensitive to FANCM depletion. In the absence of FANCM, or through synthetic inhibition of the FANCM-BTR complex, stalled forks deteriorate to form DSBs. This causes induction of break-induced telomere synthesis events to repair dysfunctional telomeres, and coincides with the production of nascent ECTR DNA species. ECTR accumulation may further exacerbate the DDR through the exhaustion of cellular RPA reserves. The consequences of DSB formation and the ensuing DDR include excessive ALT activity and loss of cell viability through G2/M stalling. Finally, inhibition of the FANCM-BTR complex represents an accessible and demonstrated therapeutic target for ALT cancers.

## Methods

**Cell culture and cell lines.** The cell lines U-2 OS, IIICF/c, HeLa, HeLa 1.2.11, HCT116, GM847, Saos-2, and HEK-293 were cultured in Dulbecco's modified Eagle's medium (DMEM) supplemented with 10% (v/v) fetal bovine serum (FBS) in a humidified incubator at 37 °C with 10% $CO_2$. Cell lines were authenticated by 16-locus short-tandem-repeat profiling and tested for mycoplasma contamination by CellBank Australia (Children's Medical Research Institute).

**RNA interference.** The following Silencer Select siRNAs were designed and synthesized by Life Technologies: FANCM1 (s33619) and FANCM2 (s33621), POLD3 (s21045), BLM (s1998), RAD51 (s11735), RAD52 (s11747) and the Silencer Select RNAi siRNA Negative Control #2 (#4390847). Cell suspensions were transfected at 20–50% confluency with Lipofectamine RNAiMAX (Life Technologies) at a final siRNA concentration of 30 nM. Culture media was changed after 48 h and cells harvested for analysis 72 h post-transfection. Knockdown efficiency was validated by western blot analysis.

**Vectors.** Empty vector and wild-type FANCM (pCMV6-Myc-DDK) constructs were obtained from Origene Technologies. Wild type FANCM and FANCM mutants were cloned into the pLenti-C-Myc-DDK-IRES-Neo backbone (Origene Technologies) by restriction enzyme subcloning of gene blocks synthesized by Integrated DNA Technologies (IDT). Lentivirus was produced by the Vector and Genome Engineering Facility (Children's Medical Research Institute). For stable overexpression, cells were transduced with lentivirus, allowed to recover for 24 h, and subject to G418 selection. Cells were maintained in G418 to ensure sustained overexpression.

**Genomic DNA extraction and purification.** Cells were harvested by trypsinization, washed in PBS, and lysed in DNA extraction buffer (100 mM Tris-HCl pH

7.6, 100 mM NaCl, 10 mM EDTA, 1% (w/v) N-lauroylsarcosine). Lysates were digested with 50 µg/ml RNase A for 20 min at room temperature, followed by digestion with 100 µg/ml proteinase K overnight at 55 °C. DNA was extracted using three rounds of phenol/chloroform/isoamyl alcohol (25:24:1) solution (Sigma Aldrich) in MaXtract High Density tubes (Qiagen). DNA from the aqueous phase was precipitated with 0.1 volume of 3 M sodium acetate pH 5.2 and 2.5 volumes of cold 100% ethanol. DNA was washed with 70% ethanol, dried, and dissolved in 10 mM Tris-HCl pH 8.0, 1 mM EDTA.

**C-circle assay.** C-circles were amplified with Phi29 polymerase using dATP, dTTP, and dGTP overnight. Products were dot blotted onto Biodyne B membranes (Pall) and pre-hybridized in PerfectHyb Plus (Sigma) for at least 30 min. γ-[$^{32}$P]-ATP-labeled telomeric C-probe (CCCTAA)$_4$ was then added and blots were hybridized overnight at 37 °C (Henson et al.[28]). Blots were washed with 0.5× SSC, 0.1% SDS three times for 5 min each then exposed to a PhosphorImager screen. Imaging was performed on the Typhoon FLA 7000 system (GE Healthcare) with a PMT of 750 V.

**Terminal restriction fragment (TRF) analysis.** Genomic DNA was digested with 4 U/µg of HinfI and RsaI overnight at 37 °C. Digested DNA was precipitated with 0.1 volume of 3 M sodium acetate pH 5.2 and 2.5 volumes of 100% ethanol. DNA was washed with 70% ethanol, dried, and dissolved in 10 mM Tris-HCl pH 7.6, 1 mM EDTA. For one-dimensional gel electrophoresis, digested DNA (2 µg) was loaded on 1% (w/v) pulse-field certified agarose (Bio-Rad) gels and separated at 6 V/cm for 12 h with an initial switch time of 1 s and a final switch time of 6 s. For two-dimensional gel electrophoresis, digested DNA (20 µg) was separated by standard gel electrophoresis in the first dimension in a 0.6% (w/v) agarose gel in 0.5× TBE at 1 V/cm for 13.5 h. Lanes were excised and run in the second dimension in a 1.1% (w/v) agarose gel containing 300 ng/ml ethidium bromide at 6 V/cm for 4 h. Gels were dried for 150 min at 50 °C and rehydrated in 2× SSC for 30 min. Gels were prehybridized in Church buffer (250 mM sodium phosphate buffer pH 7.2, 7% (w/v) SDS, 1% (w/v) BSA fraction V grade (Roche), 1 mM EDTA) for 2 h at 50 °C. Native gels were hybridized overnight with γ-[$^{32}$P]-ATP-labeled (GGGTTA)$_4$ or (CCCTAA)$_4$ oligonucleotide probes. Gels were washed three times in 0.2× SSC for 15 min at room temperature, and exposed to a PhosphorImager screen for three days. Gels were then denatured in 0.5 M NaOH, 1.5 M NaCl at 65 °C for 40 min, followed by two washes in 2× SSC. Gels were pre-hybridized for 1 h and then hybridized in Church buffer at 50 °C with γ-[$^{32}$P]-ATP-labeled (GGGTTA)$_4$ or (CCCTAA)$_4$ oligonucleotide probes overnight. Denatured gels were washed and exposed to a PhosphorImager screen overnight. Imaging was performed on the Typhoon FLA 7000 system (GE Healthcare) with a PMT of 750 V.

**Immunoblotting.** Cells were collected and lysed in RIPA buffer (50 mM Tris-HCl pH 7.6, 150 mM NaCl, 1% Nonidet P-40, 0.5% sodium deoxycholate, 0.1% SDS, 4 mM EDTA) supplemented with cOmplete Mini EDTA-free protease inhibitor cocktail (Roche). Proteins were resolved on either 3–8% Tris-Acetate or 4–12% Bis-Tris gels (Life Technologies). Proteins were transferred to Immobilon P PVDF membranes (Merck). Membranes were optionally stained with Ponceau S (Sigma Aldrich) then destained with PBST. Membranes were then blocked with either 5% skim milk or bovine serum albumin (BSA) in PBST. Blots were incubated with primary antibody (for list of antibodies, see Supplementary Table 1) at either 4 °C overnight or room temperature for 2 h. Membranes were then incubated with corresponding HRP-conjugated secondary antibodies (Dako) for 1 h at room temperature, and bands visualized using PICO, PICO PLUS, or FEMTO enhanced chemiluminescence reagents (Thermo Scientific). Uncropped blots are available in Supplementary Figs. 7 and 8.

**Immununoprecipitation (IP).** For IP, cells were lysed in IP buffer (20 mM Tris-HCl pH 7.5, 100 mM NaCl, 10% (v/v) glycerol, 0.5 mM EDTA, 1 mM DTT) supplemented with 1× mammalian protease inhibitor cocktail (Sigma-Aldrich) and 50 U/ml Benzonase (Novagen) for 2 h at 4 °C, with or without MM2 peptide variants (Mimotopes). Following lysis, NaCl concentration was increased to 250 mM, then the lysate cleared at 16,000 × g for 15 min. Supernatant was then mixed with 1 µg of α-FANCM (Abcam) or α-ER (Santa Cruz Biotechnology) and 20 µl protein G sepharose (GE Healthcare), or using α-Flag M2 agarose (Sigma Aldrich). Following 3 h of mixing at 4 °C, beads were washed 4× with IP buffer, and 1× with 50 mM NH$_4$(CO$_3$)$_2$, 0.5 mM EDTA, then eluted with 500 mM NH$_4$OH (pH 11.0), 0.5 mM EDTA. Samples were then lyophilized and resuspended in 1× LDS loading buffer (Life Technologies) prior to immunoblotting.

**Immunofluorescence (IF) and fluorescence in-situ hybridization (FISH).** Indirect IF and telomere FISH were performed on both interphase nuclei and metaphase spreads. For interphase IF experiments, cells were grown on coverslips or LabTek chamber slides (Thermo Scientific). Slides were prepared as described previously[10]. Cells on coverslips were washed twice with PBS, permeabilized with KCM buffer (120 mM KCl, 20 mM NaCl, 10 mM Tris pH 7.5, 0.1% Triton), washed again with PBST and PBS, then fixed with ice-cold 4% formaldehyde PBS solution at room temperature for 10 min. Coverslips were blocked with

antibody-dilution buffer (20 mM Tris–HCl, pH 7.5, 2% (w/v) BSA, 0.2% (v/v) fish gelatin, 150 mM NaCl, 0.1% (v/v) Triton X-100 and 0.1% (w/v) sodium azide) and 0.1 mg/ml RNase A for 30 min at 37 °C. Cells were incubated with primary antibodies (Supplementary Table 1) for 1 h at 37 °C or 2 h at room temperature, then incubated with 1:1000 dilution of appropriate Alexa Fluor conjugated secondary antibodies (Thermo Scientific). Coverslips were rinsed with PBS then fixed with 4% (v/v) formaldehyde at room temperature prior to telomere FISH. Coverslips were subjected to a graded ethanol series (75% for 2 min, 85% for 2 min, and 100% for 2 min) and allowed to air-dry. Dehydrated coverslips were overlaid with 0.3 µg/ml FAM–OO-(CCCTAA)$_3$ telomeric PNA probe (Panagene) in PNA hybridization solution (70% deionized formamide, 0.25% (v/v) NEN blocking reagent (PerkinElmer), 10 mM Tris–HCl, pH 7.5, 4 mM Na$_2$HPO$_4$, 0.5 mM citric acid, and 1.25 mM MgCl$_2$), denatured at 80 °C for 5 min, and hybridized at room temperature overnight. Coverslips were washed twice with PNA wash A (70% formamide, 10 mM Trish pH 7.5) and then PNA wash B (50 mM Tris pH 7.5, 150 mM NaCl, 0.8% Tween-20) for 5 min each. DAPI was added at 50 ng/ml to the second PNA wash B. Finally, coverslips were rinsed briefly in deionized water, air dried and mounted in DABCO (2.3% 1,4 Diazabicyclo (2.2.2) octane, 90% glycerol, 50 mM Tris pH 8.0). Microscopy images were acquired on a Zeiss Axio Imager microscope with appropriate filter sets.

**EdU detection.** Cells were pulsed with 10 µM EdU for 2 h. Cells were permeabilized, then fixed with 4% formaldehyde PBS solution. The Click-iT® Alexa Fluor 647 azide reaction was then performed according to the manufacturer's instructions, before blocking with antibody-dilution buffer and RNaseA. Telomeres were visualized by FISH using a TAMRA–OO-(CCCTAA)$_3$ telomeric PNA probe (Panagene), and PML was visualized by IF.

**Single-molecule analysis of telomeric DNA (SMAT).** Cells were labeled with 100 µM CIdU for 5 h prior to harvesting by trypsinization. Cells were embedded in low-melting agarose plugs then subjected to proteinase K digestion overnight. Plugs were dissolved with agarase (Thermo Scientific) according to the manufacturer's instructions. Molecular combing was performed using the Molecular Combing System (Genomic Vision S.A.) with a constant stretch factor of 2 kb/µm using vinyl silane coverslips (20 × 20 mm; Genomic Vision S.A.), according to the manufacturer's instructions. After combing, coverslips were dried for 4 h at 60 °C. Quality and integrity of combed DNA fibers were checked using the YoYo-1 counterstain (Molecular Probes). Coverslips were denatured for 25 min in alkali-denaturing buffer (0.2 M NaOH, 0.1% b-mercaptoethanol in 70% ethanol) and fixed by addition of 0.5% glutaraldehyde for 5 min. Telomeric DNA was visualized by hybridization with a TAMRA–OO-KKK(TTAGGG)$_3$ PNA probe (Panagene). Halogenated nucleotides were detected with a rat anti-CldU monoclonal antibody (Accurate) and Alexa Fluor 488-conjugated goat anti-rat antibody (Molecular Probes). Telomere fibers were detected on a Zeiss Axio Imager microscope with ApoTome module and analyzed with Zen software (Zeiss)[12].

**Nascent telomere analysis.** BrdU pulldown was performed as described previously, with some adaptations[67]. Cells were harvested after pulsing with 100 µM BrdU (Sigma) for 5 h. Genomic DNA (gDNA) was extracted and resuspended in 100 µl of EB buffer (Qiagen). gDNA was sheared into 100–1000 bp fragments using a Covaris M220 sonicator. Four microgram of sheared gDNA was denatured for 10 min at 95 °C, then chilled immediately. Denatured gDNA was incubated with 2 µg control mouse IgG (Millipore) or anti-BrdU antibody (BD Biosciences) in 250 µl immunoprecipitation buffer (0.0625% (v/v) Triton X-100 in PBS), rotating overnight at 4 °C. Samples were then incubated with 60 µl BSA-blocked (nuclease-free) Protein G agarose beads (Roche) at 4 °C overnight. The next day, beads were collected via centrifugation for 60 s at 13,000 rpm and washed twice with 1 ml of buffer A (20 mM HEPES-KOH, pH 8.0, 2 mM MgCl$_2$, 300 mM KCl, 1 mM EDTA, 10% (v/v) glycerol, and 0.1% (v/v) Triton X-100) and then 1 ml of TE (10 mM Tris-HCl, 1 mM EDTA, pH 8.0). Immunoprecipitated DNA was eluted from the Protein G agarose beads in 100 µl of elution buffer (50 mM NaHCO$_3$, and 1% (v/v) SDS) then purified with a QIAquick PCR Purification Kit (Qiagen) and eluted in 140 µl TE. Samples and inputs were diluted with 200 µl of 0.46 M NaOH, denatured at 95 °C for 5 min and cooled on ice, then dot blotted onto Hybond XL (GE Healthcare Life Sciences). Membranes were cross-linked with 2 × 240 mJ in a Stratalinker (Stratagene) at 254 nm, prehybridized in PerfectHyb Plus hybridization buffer (Sigma) for 30 min at 37 °C, and hybridized overnight with γ-[$^{32}$P]-ATP-labeled (TTAGGG)$_4$ telomere probe to detect the C-strand. To detect the G-strand, blots were stripped with five washes of near boiling 0.1% SDS for 20 min and reprobed with γ-[$^{32}$P]-ATP-labeled (CCCTAA)$_4$ telomere probe. Membranes were washed three times for 15 min in 2× SSC at room temperature, then exposed to a PhosphorImager screen. Imaging was performed with the array analysis function using ImageQuant software (GE Healthcare Life Sciences). Alternatively, gDNA was extracted and digested with HinfI and RsaI prior to BrdU immunoprecipitation. Immunoprecipitated DNA was then eluted from the Protein G agarose beads in 100 µl of elution buffer (50 mM NaHCO$_3$, and 1% (v/v) SDS), purified with a QIAquick PCR Purification Kit (Qiagen), and eluted in 25 µl TE. C-circle amplification and detection was carried out as described previously[28].

**Automated image analysis**. ZEN microscopy images (.czi) were processed into extended projections of z-stacks using ZEN desk 2011 software (Zeiss) and imported into Cellprofiler v2.1.1 (Carpenter et al. 2006) for analysis. The DAPI channel was used to mask individual nuclei as primary objects. Foci within each segmented nucleus were identified using an intensity-threshold based mask. Any given object was considered to be overlapping another object when at least 20% of the first object's area was enclosed within the area of a second object.

**Sister chromatid exchange (SCE) assays**. SCE assays were performed as described previously, with cells cultured in 100 μM BrdU (Sigma Aldrich), and with or without 200 nM 4-hydroxy-tamoxifen (Sigma Aldrich) for two cell cycles followed by 1 h in 0.2 μg/ml colcemid (Life Technologies). Images of mitotic spreads were captured using a Zeiss Axioplan 2 microscope, and SCEs were manually scored and blind verified. Exchanges that were obviously due to "flipping" at the centrosome were omitted from quantitation. Mitotic abnormalities were scored as described previously[68,69].

**Clonogenic assay**. Clonogenic assays were performed as described previously[70], with cells plated at 1000 cells/10 cm dish with or without 200 nM 4-hydroxytamoxifen (Sigma Aldrich), which was replenished on days 4 and 7. For cells treated with PIP-199, cells were plated at 300 cells/well in 6-well plates, and treated with 0–10 μM of drug in dimethyl sulfoxide (DMSO). Cells were left to grow for 11 days. Plates were washed with 1× PBS, then fixed and stained with 0.5% (w/v) crystal violet in 1 part acetic acid, 7 parts methanol for 2 h at room temperature. Plates were then washed with tap water and left to air dry. Colonies on each plate were counted and the plating efficiency and surviving fraction calculated.

**Live cell imaging**. Cells were seeded in Alcian-Blue coated 12-well 1.5 mm glass-bottom wells (MatTek) 24 h prior to imaging. One hundred and twenty cells and subsequent daughter cells were monitored for 48 h at 6 min intervals. The number of nuclei, interphase duration, mitotic duration (defined by duration between initial rounding up of cell to completion of cytokinesis), and mitotic outcome (normal, death during mitosis, aborted, multipolar division, and cell fusion leading to multinuclear cells), were recorded.

**Flow cytometry**. Ethanol-fixed single-cell suspensions (approximately $1 \times 10^6$ cells) were stained for DNA analysis with 2 mg/ml RNase A and 0.1 mg/ml propidium iodide (PI) in 0.25 ml PBS. Cells were incubated for 30 min at 37 °C and equilibrated at room temperature in the dark for at least 10 min. Cells were analyzed by BD FACSCanto Flow Cytometry (BD Biosciences) using an air-cooled 488 nm argon laser to excite PI. A total of 9800–10,000 stopping gate events were collected at an approximate flow rate of 200 events/s (Supplementary Fig. 9). The forward scatter (FSC, size) and side scatter (SSC, internal granularity) of each cell were recorded. To discriminate and eliminate cell debris and doublets, the pulse area (PI-A) was plotted against the pulse width (PI-W). Doublets identified as cells with 4 N DNA content and increasing pulse width were eliminated. Cell cycle population analysis was conducted with FlowJo v5 software (FlowJo) (Supplementary Fig. 9). Cell phase (G0/G1 and G2/M) gating was performed using the Dean-Jett algorithm. The percentage of cells in S-phase was calculated as the remaining percentage after G0/G1 and G2/M gating.

**Statistical analysis**. Details regarding quantitation and statistical analysis are provided in the figures and figure legends. The two-sided Student's t-test was performed on data assumed to be normally distributed, while the two-sided Mann–Whitney test was performed on data assumed to be non-normally distributed. Statistical analysis was not performed on telomere extension lengths due to the substantial difference in events between treatments. Data analysis was performed using Microsoft Excel and GraphPad Prism.

**Reporting summary**. Further information on research design is available in the Nature Research Reporting Summary linked to this article.

## Data availability

The authors declare that all data related to findings of this study are available within the article and its Supplementary Information files, or from the corresponding authors upon request. The DepMap Achilles 18Q4 public dataset used for Supplementary Fig. 5 is accessible in Figshare at https://figshare.com/articles/DepMap_Achilles_18Q4_public/7270880/2[43].

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

## Acknowledgements

Thanks to Wayne Crismani for critical reading of the manuscript. Microscopy was performed in the ACRF Telomere Analysis Center, supported by the Australian Cancer Research Foundation and the Ian Potter Foundation. Thanks to Meg Wall (Victorian Cancer Cytogenetics Service) for imaging SCE assays. A.J.D. is a Victorian Cancer Agency fellow, A.P.S. is a Cancer Institute NSW fellow, R.L. was supported by a Denise Higgins scholarship, J.J.O. received a scholarship from the Leukemia Foundation (Australia). M.L. was supported by a top-up scholarship from the Kids Cancer Alliance. This work was funded by National Health and Medical Research Council Australia (1123100 to A.J.D.), The University of Sydney (G199759 to H.A.P.), Cancer Council NSW (RG-16-09 to H.A.P.), Cancer Council Victoria (to A.J.D.), Cancer Australia (1099299 to A.P.S. and 1125750—together with National Breast Cancer Foundation—to A.J.D.) and the Victorian Government OIS Program.

## Author contributions

R.L., J.J.O., A.P.S., J.A.M.A., C.B.N., C.G.T., M.L. performed experiments and analyzed the data. H.A.P., A.J.D., and R.R.R. supervised the study and wrote the paper. All authors read and edited the paper.

## Additional information

**Competing interests:** The authors declare no competing interests.

