## [Peer Review File · Nature Communications]

Reviewers' comments:

Reviewer #1 (Remarks to the Author):

Lu et al. examines the consequences of FANCM depletion in the context of telomere regulation by the ALT mechanism. The authors show that depletion of FANCM leads to increased frequency of ALT associated markers in particular ALT specific foci and DNA species termed C-circles. The authors then link this to unrestrained BLM-POLD3 mediated DNA synthesis. They also conduct a structure function analysis and convincingly demonstrate that these effects are due to the translocase function of FANCM but are independent of the canonical function of FANCM within the FA core complex.

Overall, the manuscript is well written – some statements raise questions – and the experiments are well done and thorough. The structure functional analysis of FANCM is strengthens the study. The main weakness of the study is that despite the evident large effect on ALT – there is no change in telomere length, yet the cells appear sensitive and die. I have some concerns relating to specificity of the phenotype and others that if addressed would enhance the paper.

1. Most of the experiments are conducted in U2OS cells. The other ALT line is H1ICF/c – in which the effects are less obvious. U2OS cells have an allele of p53 so the G2/M checkpoint could be intact, at least partially. Certainly this could explain some of the results. The results should be confirmed in additional lines or the G2/M checkpoint in U2OS cells should be abrogated by p53 protein removal (with viruses or siRNAs for example).

2. The authors claim that these effects are specifically attributable to telomere dysfunction but the effects of FANCM on the rest of the genome are not examined.

o Is there an increase in DDR signaling – globally?

o Is there a global effect on DNA recombination/replication (as Figure S5d appears to suggest is the case)?

o Is there a global effect in telomerase positive cells?

3. The direct association of FANCM with telomeres is not shown? This could help address questions of indirect/direct effects. Could this not be addressed with expression of the myc-tagged constructs or antibody?

4. Despite the evident effect on CCs and APBs – no effect is seen on telomere length. Therefore, the significance of the results are unclear, especially the SMAT results presented in Figure 2. The statement is made in the text (Pg.6) and discussion that there is a reliance on both RAD51 and RAD52 with respect to the DNA synthesis results. As the authors are undoubtedly aware – this is somewhat contrary to published results.

o In Figure 2a, the change on CCs is not statistically significant and the effect of RAD51 on telomere extension by SMAT is ~1%.

o If there is no change on telomere length after FANCM depletion– then what is the SMAT actually measuring? Is it accurate to name this telomere extension?

o Knockdown of FANCM shown in figure S2a also reduces the levels of BLM, POLD3 and RAD51.

This is clear to my eye, and not beyond the realm of possibility. If confirmed should be quantified and considered. Also, there are multiple bands for RAD52 (I have not seen this in other manuscripts concerning RAD52). I am sure the authors can look into this, I might be wrong – but it could be important to clarify this.

5. The experiments with PIP199 should be conducted in other ALT+ and TEL+ cell lines to support the conclusion.

6. What is the inference drawn from the intensity of foci? Is this meant to imply changes in telomere clustering or that there is simply more telomeric DNA in FANCM cells?

Reviewer #2 (Remarks to the Author):

ALT telomeres are known to experience chronic DNA damage that may drive recombination-based telomere elongation. In previous studies, the authors of this manuscript have implicated break induced replication (BIR), a repair pathway for one ended double strand breaks in the ALT mechanism. While the general understanding of the ALT mechanism continues to evolve, the key factors involved in regulating ALT activity have not been fully fleshed out. In this manuscript the authors identify FANCM as a critical regulator of ALT activity. Specifically, FANCM depletion exacerbates ALT phenotype suggesting a role for FANCM in the suppression of ALT activity. In addition, the authors identify the MM2 domain of FANCM as the region responsible for the interaction with the BTR (BLM-TOP3a-RMI) complex and that this interaction is critical for the suppression of ALT activity. Finally, the authors demonstrate that synthetic disruption of the FANCM-BTR complex mimics FANCM depletion and results in the loss of ALT cell viability. Overall, the authors clearly demonstrate a critical role for FANCM in regulating replication stress specifically at ALT telomeres. The manuscript is well written and the central conclusions are well supported by the data provided. The data are robust and there is attention to rigor and statistical analysis.

Minor Concerns:

1. In figure 1b, S1b and S2b, there was a striking increase in low molecular weight single-stranded (ss) C-rich telomeric DNA with FANCM depletion under native, but not under denaturing conditions. The authors did not discuss this difference perhaps this is simply a difference in exposure time between the 2 conditions?
2. In figure S2a, there appears to be a partial depletion of PolD3, BLM and RAD51 in siFANCM2 alone condition. Is it a consequence of FANCM depletion, off target effects of siFANCM2, or a fluke of that particular WB? Are the robust phenotypes of FANCM depletion by siFANCM2 a combined effect of the consequential reduction in the expression of PolD3, BLM and RAD51?
3. In Figure 3d the DNA fiber experiments are elegant, but could use some clarification as to how the authors analyze the terminal ends of telomeres without labeling parental DNA strands. Couldn't it just be a fragmented fiber, given that these FANCM deficient cells have such elevated levels of replication stress at the telomeres anyway? The authors only show a single fiber, but perhaps in a field of fibers stained with YOYO you could see that in fact the fibers as a whole are not fragmented? Or perhaps unnecessary, but if you incubate cells with CldU for 24 hours and then treat cells with your IdU pulse, you could label all of the DNA with CldU and only the replicating DNA with IdU. When you stain you could use the antibodies for CldU and IdU and a biotinylated probe for telomeres. This would allow you to detect the full length of the fiber (CldU), the terminal end being elongated (IdU), and that this is occurring at telomeres (probe)? The fibers presented look great, just curious how the authors are confident in these terminal events.
4. In Figure 4, the data almost argue that re-expression of FANCM represses ALT phenotypes in ALT cells. Is this complete? Do you see phenotypes similar to non-ALT cells following re-expression? What happens to ALT cells that over-express FANCM over time?
5. If you bypass G2/M arrest (ATR inhibition?) in FANCM deficient cells do they undergo mitotic catastrophe? Or arrest with massive damage in G1?
6. The authors state that disruption of the FANCM-BTR complex leads to a decrease in viability. Is it really decreased viability? Or is this consistent with the previous result that the cells are arresting in G2 and therefore truly have defects in proliferation? If it's viability, it might be nice to analyze the Project Achilles database? There are at least a handful of ALT positive cell lines in here that could be compared to non-ALT cell lines in terms of viability following FANCM shRNA and/or CRISPR knockout. This might complement the authors statements about the difficulty in obtaining CRISPR knockout clones?

Reviewer #3 (Remarks to the Author):

In the manuscript entitled "The FANCM_BLM-TOP3A-RMI complex suppresses alternative lengthening of telomeres (ALT)", Lu et al. investigate the effects of FANCM depletion in cells that

maintain their telomeres through recombination (ALT). They observed an exaggeration of the ALT phenotype following FANCM depletion, with regards to C-circle accumulation, frequency of telomere extension, TIF formation and the prevalence of APBs. These exaggerated defects were dependent on FLM and POLD3 and culminated in a growth defect, with cells accumulating in the G2 phase of the cell cycle.

Importantly, the authors demonstrated that it is the specific disruption of the BTR-FANCM complex which leads to lethality in ALT cells (three different cell lines tested). This was demonstrated using a 28 a.a. peptide of the MM2 domain of FANCM which could saturate the binding site of the BTR complex and specifically prevent the BTR-FANCM interaction. In addition, the inhibitor PIP-199, which disrupts the MM2-RMI interaction was also able to specifically suppress the growth of ALT cells. Inhibition of the complex formation by the small molecule inhibitor PIP-199 is a promising strategy for ALT tumour treatment.

This is a very important study that will have a broad readership beyond the telomere field. The manuscript is well laid out and clearly written. I support its publication in Nat. Comm. Nonetheless a few additional experiments/controls could strengthen the manuscript to an extent.

As a control, it would be nice to see whether the inhibition of FANCM is toxic in telomerase positive cells that have been deleted for ATRX to rule out the notion that the effect seen in U2OS cells are not due to a synthetic lethal effect with loss of ATRX, which may be independent of telomere status.

This manuscript has clearly established that FANCM is required for the optimal growth of ALT cell lines. However, what about the establishment of ALT? The authors have shown that there are no effects in telomerase positive HeLa cells, but what about HeLa cells where telomerase has been inhibited or deleted. Does the inhibition of FANCM in this scenario drive ALT like phenotypes. Does FANCM have to be down-regulated for cells to transition to ALT? Can the siASF1 in HeLa-LT cells be used here combined with FANCM overexpression. I am hoping that perhaps one of these approaches can be used to address this question.

Minor text changes that should be addressed:

-p.3, top: 'FANCM is an integral factor in the stabilization of stalled replication forks'. Here Luke-Glaser et al. EMBO Journal 2010 should be cited as well. Problems in fork restart were shown in this manuscript by DNA fibre analysis.

-in the first section of the results section (last paragraph) there are some mix ups with figure labeling (i.e. sometimes supplemental figures are referred to instead of main figures) please check carefully

-In Figure 4 the "f" should be removed because "e" and "f" are both described within "e" in the figure legend

-p.6: again, please check the figures you have referred to (Fig. S3a and S2b is not correct as far as I can tell)

-p.7: 'MHF1/2 heterotetramer interface' instead of 'MH1/2 heterotetramer interface'.

-p.14: 'Specifically, FANCM functions in a BTR-dependent, FA-core complex-independent manner, to remodel stalled replication forks that arise spontaneously within telomeres.' We presume to remodel replication forks, although the resolution of BIR strand invasion cannot be ruled out. Yeast Mph1 is able to perform this function in vitro. Moreover, FANCM can resolve R-loops. Perhaps the remodeling fork statement could be slightly toned down.

Reviewer #1

Lu et al. examines the consequences of FANCM depletion in the context of telomere regulation by the ALT mechanism. The authors show that depletion of FANCM leads to increased frequency of ALT associated markers in particular ALT specific foci and DNA species termed C-circles. The authors then link this to unrestrained BLM-POLD3 mediated DNA synthesis. They also conduct a structure function analysis and convincingly demonstrate that these effects are due to the translocase function of FANCM but are independent of the canonical function of FANCM within the FA core complex.

Overall, the manuscript is well written – some statements raise questions – and the experiments are well done and thorough. The structure functional analysis of FANCM is strengthens the study. The main weakness of the study is that despite the evident large effect on ALT – there is no change in telomere length, yet the cells appear sensitive and die. I have some concerns relating to specificity of the phenotype and others that if addressed would enhance the paper.

1. Most of the experiments are conducted in U2OS cells. The other ALT line is IICF/c – in which the effects are less obvious. U2OS cells have an allele of p53 so the G2/M checkpoint could be intact, at least partially. Certainly this could explain some of the results. The results should be confirmed in additional lines or the G2/M checkpoint in U2OS cells should be abrogated by p53 protein removal (with viruses or siRNAs for example).

We have now confirmed our results in three additional cell lines. These include GM847 (ALT, SV40 transformed), Saos-2 (ALT, p53 null) and HCT116 (telomerase, p53 wild-type). Specifically, C-circles increased approximately 15-20-fold in GM847 and approximately 10-fold in Saos-2 following FANCM depletion. While the number of APBs increased significantly in U-2 OS and IICF/c cells, the number of APBs did not increase in GM847 and Saos-2 cells. Interestingly, telomere intensity within APBs increased significantly in all ALT cell lines analysed following FANCM depletion, consistent with increased telomere clustering. No induction of C-circles or APBs was observed in HCT116 cells in response to FANCM depletion. These data are included in Supplementary Fig. 1a, 1d, 1e and 1f of the revised manuscript, and the text has been modified to include, clarify and interpret these findings.

2. The authors claim that these effects are specifically attributable to telomere dysfunction but the effects of FANCM on the rest of the genome are not examined.

o Is there an increase in DDR signaling – globally?

It has previously been shown that FANCM depletion causes an increase in global DDR signalling (Blackford et al., 2012). Consistent with this finding, we observed an increase in global 53BP1 foci in interphase cells in both U-2 OS and HeLa cells following FANCM depletion. These data have been included in Supplementary Fig. 1b, and the following text has been added to the Results section of the revised manuscript “In contrast to the ALT-specific induction of telomere dysfunction, FANCM depletion induced comparable levels of global DDR signalling in both U-2 OS (ALT) and HeLa (telomerase-positive) cell lines (Supplementary Fig. 1b)”, and “...indicative of the induction of ALT phenotypes being attributable to ALT-specific telomere dysfunction.”

o Is there a global effect on DNA recombination/replication (as Figure S5d appears to suggest is the case)?

o Is there a global effect in telomerase positive cells?

It has previously been shown that FANCM depletion has a global effect on DNA recombination in both ALT and telomerase-positive cells, evident by an increase in sister chromatid exchange events following FANCM depletion (Deans and West, 2009). This is consistent with our findings in Supplementary Fig. 6. To address this point, we have cited this paper and added the following sentence to the Results section “It has previously been shown that FANCM depletion results in increased SCE formation.”

3. The direct association of FANCM with telomeres is not shown? This could help address questions of indirect/direct effects. Could this not be addressed with expression of the myc-tagged constructs or antibody?

The direct association between FANCM and telomeres has been demonstrated previously (Pan et al., 2017) by combined IF and telomere FISH. In our study, we were unable to obtain robust IF data demonstrating a direct association between FANCM and telomeres, even using the myc-tagged constructs. We were able to identify rare colocalizations, but we did not feel the data were of sufficient quality to include in the manuscript. We believe that this is due to the interaction between FANCM and telomeres being dynamic, and FANCM not aggregating to form visible foci, compounded by the sub-optimal antibodies available.

Direct association between FANCM and telomeres can also be demonstrated by ChIP using an anti-FANCM antibody. This has been done in the accompanying manuscript (Pentz et al., *Nat Commun*), in which direct association between FANCM and telomeric DNA is demonstrated in U-2 OS, VA13, HOS, HeLa and HT1080 cell lines. This interaction is greatest in the ALT cell lines compared to the telomerase-positive cell lines. Upon consultation with the *Nature Communications* Editor, and to avoid costly and redundant experiments, we have referenced this accompanying manuscript.

We have added the following sentence “It has been shown elsewhere that FANCM directly associates with telomeric DNA” and cited both papers as support for a direct interaction between FANCM and telomeres.

4. Despite the evident effect on CCs and APBs – no effect is seen on telomere length. Therefore, the significance of the results are unclear, especially the SMAT results presented in Figure 2. The statement is made in the text (Pg.6) and discussion that there is a reliance on both RAD51 and RAD52 with respect to the DNA synthesis results. As the authors are undoubtedly aware – this is somewhat contrary to published results.

Our data suggest that ALT activity is partially dependent on RAD51 and RAD52. This is consistent with a recent publication that demonstrates ALT is mediated by two distinct pathways, one dependent on RAD52 and the other independent of RAD52 (Zhang et al., 2019). We have now cited this paper in the Discussion section, as support for our findings.

o In Figure 2a, the change on CCs is not statistically significant and the effect of RAD51 on telomere extension by SMAT is ~1%.

The text has been amended to clarify that (i) the increase in CCs following FANCM depletion was only partially dependent on RAD51 and RAD52, and (ii) the increase in telomere extension events following FANCM depletion was predominantly dependent on POLD3, BLM and RAD52.

o If there is no change on telomere length after FANCM depletion– then what is the SMAT actually measuring? Is it accurate to name this telomere extension?

We have used SMAT to measure two specific parameters that contribute to telomere extension: (i) the frequency of extended telomeres as a proportion of total telomere fibres (identified as fibres with terminal CldU incorporation), and (ii) the length of the extended portion of the telomere (measured by the length of the terminal CldU-incorporated portion of the telomere) (Sobinoff et al., 2017). Our data support FANCM depletion promoting replication fork collapse and the initiation of extension events at broken forks; however, the length of the extension events does not change. Both factors contribute to overall telomere extension, and the SMAT assay is able to delineate the two.

The finding that there is no change to overall telomere length may also be attributable to the short-term timeline of these experiments. We were unable to achieve FANCM knockout or stable FANCM knockdown, meaning that all knockdown experiments are conducted transiently at 72 hours post siRNA transfection. It is possible that telomere length may be affected, but that the changes are undetectable by TRF analysis during the short timeline. To clarify this point, the following text has been added to the Discussion section of the revised manuscript “No changes in mean telomere length were observed during transient FANCM depletion; however, the short timeline may preclude the detection of telomere length changes by TRF analysis. These data also suggest that ECTRs are a by-product, rather than a substrate for break-induced telomere synthesis, and that ECTR generation may counteract telomere extension.”

o Knockdown of FANCM shown in figure S2a also reduces the levels of BLM, POLD3 and RAD51. This is clear to my eye, and not beyond the realm of possibility. If confirmed should be quantified and considered. Also, there are multiple bands for RAD52 (I have not seen this in other manuscripts

concerning RAD52). I am sure the authors can look into this, I might be wrong – but it could be important to clarify this.

We appreciate this observation, which we had missed. We have now confirmed that FANCM depletion causes a decrease in the levels of POLD3, BLM and RAD51. We have quantitated the results, and determined that the decrease is significant for RAD51. This is an important observation, which we have now added to the revised manuscript. Western blot quantitation is included in Supplementary Fig. 2b of the revised manuscript.

The robust phenotypes caused by FANCM depletion are unlikely to be a combined effect of the consequential reduction in the expression of POLD3, BLM and RAD51, as we have previously shown that when these proteins are independently depleted the effects to ALT activity are subtle or antagonistic. To address this point, the following text has been added to the Results section of the revised manuscript “It is unlikely that this coregulation contributes to the exacerbated ALT phenotype observed, as independent depletion of POLD3, BLM or RAD51 causes subtle or antagonistic effects to ALT activity, compared to that seen with FANCM depletion.”

We routinely observe multiple bands with our RAD52 antibody. This is consistent with a previous publication that uses the same antibody (Mijic et al., 2017).

5. The experiments with PIP199 should be conducted in other ALT+ and TEL+ cell lines to support the conclusion.

We have extended the PIP-199 experiments to include GM847 (ALT), Saos-2 (ALT) and HCT116 (telomerase) cell lines. The results are consistent, demonstrating increased sensitivity to PIP-199 in the ALT cell lines compared to the telomerase-positive cell lines, and an ALT-specific increase in C-circles. These data have been included in Fig. 5f and 5g of the revised manuscript.

6. What is the inference drawn from the intensity of foci? Is this meant to imply changes in telomere clustering or that there is simply more telomeric DNA in FANCM cells?

The inference is that both telomere clustering and an increase in telomeric DNA contribute to the increased telomere intensity in APB foci following FANCM depletion. Specifically, as demonstrated in Fig. 2 and described in the associated Results and Discussion sections, the increased intensity of telomeric signal is consistent with the accumulation of newly synthesised, and predominantly extrachromosomal, telomeric DNA following FANCM depletion. In addition, the new data presented in Supplementary Fig. 1e and 1f of the revised manuscript show that, while the number of APBs/cell increases only in 2/4 ALT cell lines, telomere intensity in APB foci increases substantially in all four ALT cell lines, indicative of telomere clustering. The interpretation of these data has been clarified in the Results section of the revised manuscript.

Reviewer #2

ALT telomeres are known to experience chronic DNA damage that may drive recombination-based telomere elongation. In previous studies, the authors of this manuscript have implicated break induced replication (BIR), a repair pathway for one ended double strand breaks in the ALT mechanism. While the general understanding of the ALT mechanism continues to evolve, the key factors involved in regulating ALT activity have not been fully fleshed out. In this manuscript the authors identify FANCM as a critical regulator of ALT activity. Specifically, FANCM depletion exacerbates ALT phenotype suggesting a role for FANCM in the suppression of ALT activity. In addition, the authors identify the MM2 domain of FANCM as the region responsible for the interaction with the BTR (BLM-TOP3a-RMI) complex and that this interaction is critical for the suppression of ALT activity. Finally, the authors demonstrate that synthetic disruption of the FANCM-BTR complex mimics FANCM depletion and results in the loss of ALT cell viability. Overall, the authors clearly demonstrate a critical role for FANCM in regulating replication stress specifically at ALT telomeres. The manuscript is well written and the central conclusions are well supported by the data provided. The data are robust and there is attention to rigor and statistical analysis.

Minor Concerns:

1. In figure 1b, S1b and S2b, there was a striking increase in low molecular weight single-stranded (ss) C-rich telomeric DNA with FANCM depletion under native, but not under denaturing conditions. The authors did not discuss this difference perhaps this is simply a difference in exposure time between the 2 conditions?

This is an exposure issue. There is substantially more double-stranded telomeric DNA compared to single-stranded telomeric DNA in cells. Consequently, the signal on the native gels is much lower than that on the denatured gels. The native gels were exposed to phosphor screens for approximately six times as long as the denatured gels. Consistent with this, the low molecular weight single-stranded C-rich telomeric DNA observed following FANCM depletion can still be seen, albeit more faintly, on the denatured gels.

2. In figure S2a, there appears to be a partial depletion of PolD3, BLM and RAD51 in siFANCM2 alone condition. Is it a consequence of FANCM depletion, off target effects of siFANCM2, or a fluke of that particular WB? Are the robust phenotypes of FANCM depletion by siFANCM2 a combined effect of the consequential reduction in the expression of PolD3, BLM and RAD51?

Please see response to reviewer #1, point 4.

3. In Figure 3d the DNA fiber experiments are elegant, but could use some clarification as to how the authors analyze the terminal ends of telomeres without labeling parental DNA strands. Couldn't it just be a fragmented fiber, given that these FANCM deficient cells have such elevated levels of replication stress at the telomeres anyway? The authors only show a single fiber, but perhaps in a field of fibers stained with YOYO you could see that in fact the fibers as a whole are not fragmented? Or perhaps unnecessary, but if you incubate cells with CldU for 24 hours and then treat cells with your IdU pulse, you could label all of the DNA with CldU and only the replicating DNA with IdU. When you stain you could use the antibodies for CldU and IdU and a biotinylated probe for telomeres. This would allow you to detect the full length of the fiber (CldU), the terminal end being elongated (IdU), and that this is occurring at telomeres (probe)? The fibers presented look great, just curious how the authors are confident in these terminal events.

As described in our previous publication (Sobinoff et al., 2017), we categorised telomere fibres as (i) non-replicating (no CldU incorporation), (ii) replicating (CldU incorporation in the telomere-adjacent region extending into the telomere), and (iii) telomere extension events (CldU incorporation limited to one end of the telomere fibre with no overlap with the telomere-adjacent region). Unfortunately, the YoYo-1 counterstain is not compatible with the denaturation protocol required for telomere FISH. However, in order to minimise the potential contribution of fragmented fibres, we routinely counterstain a control coverslip for each sample preparation with YoYo-1, to determine the quality and integrity of the combed fibres prior to experimental staining and analysis (Rebuttal Fig. 1A). We have included this important technical detail in the Methods section of the revised manuscript.

In our hands, the telomere fibre assay works well; however, we are always interested in technical improvements. In response to the reviewer's suggestion, we have attempted CldU/IdU staining in conjunction with using a biotinylated probe to detect the telomeric DNA (Rebuttal Fig. 1B). Results were

Rebuttal Figure 1. Experimental validation of telomere fibre analysis in U-2 OS cells. (A) Example image of DNA fibre integrity visualised with YoYo-1 (green). (B) Example image of telomere extension fibre (red) following 19 hour CldU (blue) and 5 hour IdU (green) incorporation. (C) Quantitation of the number and length of telomere extension events following FANCM depletion.

consistent with those obtained using the single CldU pulse (Rebuttal Fig. 1C), supporting this modification as a future improvement for the detection of full-length fibres.

4. In Figure 4, the data almost argue that re-expression of FANCM represses ALT phenotypes in ALT cells. Is this complete? Do you see phenotypes similar to non-ALT cells following re-expression? What happens to ALT cells that over-express FANCM over time?

The repression of ALT phenotypes following FANCM re-expression is not complete. Specifically, FANCM re-expression resulted in a significant decrease in meta-TIFs, fragile telomeres and C-circles, but negligible changes in telomere extension events, telomere length and APBs (Fig. 4 and Supplementary Fig. 3). To address the reviewer's curiosity, we have cultured U-2 OS cells stably overexpressing FANCM wild-type and mutant constructs for 35 population doublings. Expression was maintained during this time-course (Rebuttal Fig. 2A). There was no overall change in telomere length (Rebuttal Fig. 2B). C-circles increased slightly at later timepoints following the initial repression observed in response to FANCM overexpression (Rebuttal Fig. 2C). We rationalise that FANCM re-expression alleviates telomeric replication stress, but that the changes to telomere extension by break-induced replication are insufficient to influence telomere length or the telomere maintenance mechanism. For simplicity we have not included these data in the manuscript, but have clarified our interpretation of the results of FANCM re-expression in the Results and Discussion sections.

Rebuttal Figure 2. Long-term overexpression of FANCM constructs does not impact telomere maintenance mechanism. (A) Western immunoblotting of stable wild-type FANCM (FANCM+) or FANCM mutant overexpression in U-2 OS cells at early (PD0) and late (PD35) population doublings. (B) TRF analysis of U-2 OS cells overexpressing wild-type (FANCM+) or FANCM mutants at early (PD0) and late (PD35) population doublings. (C) Representative dot blots and quantitation of C-circle assays in U-2 OS cells stably overexpressing wild-type (FANCM+) or FANCM domain mutants at early (PD0) and late (PD35) population doublings. C-circles were normalized to the vector control.

5. If you bypass G2/M arrest (ATR inhibition?) in FANCM deficient cells do they undergo mitotic catastrophe? Or arrest with massive damage in G1?

To address this point, we used the ATR inhibitor VE822 to bypass G2/M arrest in cells with or without FANCM depletion. We observed that cells did not undergo mitotic catastrophe, but did arrest in G1 (Rebuttal Fig 3A and 3B). There was, however, no obvious exacerbation of damage following ATR inhibition and FANCM depletion (Rebuttal Fig. 3C). This requires further investigation. For simplicity, we have not included these data in the revised manuscript.

Rebuttal Figure 3. ATR inhibition in FANCM depleted U-2 OS cells resulted in G1 arrest. (A) Mitotic outcome was not affected by ATR inhibition. (B) ATR inhibition in the context of FANCM depletion resulted in G1 arrest. (C) Western immunoblotting of γ H2AX.

6. The authors state that disruption of the FANCM-BTR complex leads to a decrease in viability. Is it really decreased viability? Or is this consistent with the previous result that the cells are arresting in G2 and therefore truly have defects in proliferation? If it's viability, it might be nice to analyze the Project Achilles database? There are at least a handful of ALT positive cell lines in here that could be compared to non-ALT cell lines in terms of viability following FANCM shRNA and/or CRISPR knockout. This might complement the authors statements about the difficulty in obtaining CRISPR knockout clones?

Thank you for this suggestion. We have now analysed the Project Achilles database. We identified five ALT cell lines (CAL-78, G-292, Saos-2, SK-N-FI and U-2 OS) across the panel. 4/5 of these cell lines had a gene dependency score that clustered around -1, indicative of FANCM being essential for cell viability in these cell lines. We have included these data in Supplementary Fig. 5 and added an additional paragraph in the "Synthetic disruption of FANCM-BTR complex formation inhibits ALT cell viability" Results section of the revised manuscript.

Reviewer #3

In the manuscript entitled "The FANCM_BLM-TOP3A-RMI complex suppresses alternative lengthening of telomeres (ALT)", Lu et al. investigate the effects of FANCM depletion in cells that maintain their telomeres through recombination (ALT). They observed an exaggeration of the ALT phenotype following FANCM depletion, with regards to C-circle accumulation, frequency of telomere extension, TIF formation and the prevalence of APBs. These exaggerated defects were dependent on FLM and POLD3 and culminated in a growth defect, with cells accumulating in the G2 phase of the cell cycle. Importantly, the authors demonstrated that it is the specific disruption of the BTR-FANCM complex which leads to lethality in ALT cells (three different cell lines tested). This was demonstrated using a 28 a.a.

peptide of the MM2 domain of FANCM which could saturate the binding site of the BTR complex and specifically prevent the BTR-FANCM interaction. In addition, the inhibitor PIP-199, which disrupts the MM2-RMI interaction was also able to specifically suppress the growth of ALT cells. Inhibition of the complex formation by the small molecule inhibitor PIP-199 is a promising strategy for ALT tumour treatment.

This is a very important study that will have a broad readership beyond the telomere field. The manuscript is well laid out and clearly written. I support its publication in Nat. Comm. Nonetheless a few additional experiments/controls could strengthen the manuscript to an extent.

1. As a control, it would be nice to see whether the inhibition of FANCM is toxic in telomerase positive cells that have been deleted for ATRX to rule out the notion that the effect seen in U2OS cells are not due to a synthetic lethal effect with loss of ATRX, which may be independent of telomere status.

To address this point, we conducted clonogenic survival assays on telomerase positive HCT116 wild-type and ATRX knockout clones (Napier et al., 2015). We found no change in cellular survival between HCT116 wild-type and HCT116 ATRX knockout cells following FANCM depletion (Rebuttal Fig 4), indicative of the effects seen in U-2 OS cells not being attributable to synthetic lethality. We have not included these negative data in the manuscript.

Rebuttal Figure 4. Deletion of ATRX does not impact cellular survival following FANCM depletion. Quantitation of the surviving fraction of colonies for HCT116 wild-type and HCT116 ATRX knockout cells with or without FANCM depletion.

2. This manuscript has clearly established that FANCM is required for the optimal growth of ALT cell lines. However, what about the establishment of ALT? The authors have shown that there are no effects in telomerase positive HeLa cells, but what about HeLa cells where telomerase has been inhibited or deleted. Does the inhibition of FANCM in this scenario drive ALT like phenotypes? Does FANCM have to be down-regulated for cells to transition to ALT? Can the siASF1 in HeLa-LT cells be used here combined with FANCM overexpression. I am hoping that perhaps one of these approaches can be used to address this question.

This is an interesting possibility. To address the reviewer's comment, we co-depleted FANCM and hTERT in HeLa cells, and observed no induction of C-circles (Rebuttal Fig. 5). We have not included these negative data in the manuscript.

As part of a separate study, we have depleted FANCM in matched loxIMVI telomerase-positive and telomerase-null (hTR knock-out) cells, and similarly identified no change in C-circles. Overall, these data suggest that FANCM-depletion in the context of telomerase inhibition is insufficient to activate ALT.

Rebuttal Figure 5. Depletion of FANCM in the context of telomerase inhibition is insufficient to induce ALT phenotypes. Representative dot blots of C-circle assays from HeLa cells following FANCM depletion or co-depletion of FANCM and hTERT. U-2 OS was included as a positive control.

Minor text changes that should be addressed:

-p.3, top: 'FANCM is an integral factor in the stabilization of stalled replication forks'. Here Luke-Glaser et al. EMBO Journal 2010 should be cited as well. Problems in fork restart were shown in this manuscript by DNA fibre analysis.

This citation has been included.

-in the first section of the results section (last paragraph) there are some mix ups with figure labeling (i.e. sometimes supplemental figures are referred to instead of main figures) please check carefully

This has been corrected.

-In Figure 4 the "f" should be removed because "e" and "f" are both described within "e" in the figure legend

This has been corrected.

-p.6: again, please check the figures you have referred to (Fig. S3a and S2b is not correct as far as I can tell)

This has been corrected.

-p.7: 'MHF1/2 heterotetramer interface' instead of 'MH1/2 heterotetramer interface'.

This has been corrected.

-p.14: 'Specifically, FANCM functions in a BTR-dependent, FA-core complex-independent manner, to remodel stalled replication forks that arise spontaneously within telomeres.' We presume to remodel replication forks, although the resolution of BIR strand invasion cannot be ruled out. Yeast Mph1 is able to perform this function in vitro. Moreover, FANCM can resolve R-loops. Perhaps the remodeling fork statement could be slightly toned down.

This sentence has been toned down.

REVIEWERS' COMMENTS:

Reviewer #1 (Remarks to the Author):

The authors have done a very good job in addressing the issues that were raised. Thank you.

Reviewer #3 (Remarks to the Author):

I appreciate all revisions that the authors have performed. My concerns were all addressed and I fully support the publication of this very nice manuscript.